# Modeling river water temperature with limiting forcing data: air2stream v1.0.0, machine learning and multiple regression

Manuel Almeida[1], Pedro Coelho[2]

[1,2] Faculdade de Ciências e Tecnologia, Universidade Nova de Lisboa, Mare – Centro de Ciências do Mar e do Ambiente, Lisboa, 2825 - 516, Portugal.

*Correspondence to*: Manuel Almeida (mcvta@fct.unl.pt)

## Abstract

The prediction of river water temperature is of key importance in the field of environmental science. Water temperature datasets for low order rivers are often in short supply, leaving environmental modelers with the challenge of extracting as much information as possible from existing datasets, usually without the use of physically based models, due to the significant amount of data required (e.g., river morphology, degree of shading, wind velocity). Therefore, the main goal of this study is to identify a suitable modeling solution for the prediction of river water temperature given the scarcity of the forcing datasets. In this study, five models, forced with the meteorological datasets obtained from the fifth-generation atmospheric reanalysis, ERA5-Land, are used to predict the water temperature of 83 rivers (with 98% missing data): three machine-learning algorithms (Random Forest, Artificial Neural Network and Support Vector Regression), the hybrid Air2stream model with all available parameterizations and a Multiple Regression. With the exception of Air2stream, which was forced with mean daily air temperature and discharge, all other models were forced with mean, maximum, and minimum daily air temperature, mean daily total radiation (shortwave), mean daily discharge, month of the year and day of the year. The machine learning hyperparameters were optimized with a Tree-structured Parzen Estimators algorithm and the results of each model are presented as an ensemble of 12 individual optimized model runs. Additionally, an over/undersampling technique was used to generate 100 synthetic training datasets from 12 raw training datasets. These datasets were then used to improve the prediction efficiency of the best model. In general terms, the results of the study demonstrate the vital importance of hyperparameter optimization and suggest that, from a practical modeling perspective, when the number of predictor variables and observed river water temperature values are limited, the application of all the models considered in this study is crucial. Basically, all the models tested proved to be the best for at least one station. The Root Mean Square Error (RMSE) and the Nash-Sutcliffe efficiency (NSE) values obtained for the ensemble of all model results was 2.75ºC ± 1.00 and 0.56ºC ± 0.48, respectively. The model that performed the best overall was Random Forest (annual mean - RMSE: 3.18ºC ± 1.06; NSE: 0.52 ± 0.23). With the application of the over/undersampling technique, the RMSE values obtained with the Random Forest model were reduced from 0.00% to 21.89% (μ=8.57%; σ=8.21%) and the NSE values increased from 1.1% to 217.0% (μ=40%; σ=63%). These results suggest that the solution proposed has the potential to significantly improve the modeling of water temperature in rivers with machine learning methods, as well as providing increased scope

for its application to larger training datasets and the prediction of other types of dependent variables. The results also revealed the existence of a logarithmic correlation among the RMSE between the observed and predicted river water temperature and the watershed time of concentration. The RMSE increases by an average of 0.1 ℃ with a one-hour increase in the watershed time of concentration (watershed area: $\mu= 106$ km$^2$; $\sigma=153$).

## 1 Introduction

Water temperature (WT) is recognized as a key parameter in aquatic systems due to its influence on water quality (e.g., chemical reaction rate; oxygen solubility) and the distribution and growth rate of aquatic organisms (e.g., primary production; fish growth and habitat) (Smith, 1972; Webb, 2003; Caissie, 2006). As such, the accurate prediction and assessment of river WT is a crucial part of many earth science applications. The thermal dynamics in rivers is quite complex as it depends on an array of physical and chemical factors (Smith and Lavis, 1975; Jeppesen and Iversen, 1987). River WT follows a seasonal and a diurnal cycle, driven by heat input and losses at the boundary conditions of a river section (upstream and downstream transfer; air-water and sediment-water interface; lateral contribution from tributaries and groundwater) under specific meteorological and hydrological conditions (Walling and Webb, 1993; Wetzel, 2001). The complexity of river WT estimation is often more pronounced for sub-daily temporal and spatial scales (Toffolon and Piccolroaz, 2015) and it is, therefore, common practice to average out sub-daily effects and to consider a daily discretization for modeling purposes. This assumption can have a significant impact on lake/reservoir water quality modeling results, namely, when lake/reservoir inflows are large. The fall and spring turnover onset, stratification strength/length and the overall heat budget can be affected, therefore, some caution is needed regarding this type of approach. Air temperature correlates with the equilibrium temperature of a river and is, therefore, frequently used as the independent variable; hence, it is not unusual to find a strong linear correlation between daily air temperature and stream and river WT with a time lag (Smith, 1981; Crisp and Howson, 1982). The existing body of literature includes a number of examples of the successful implementation of linear regression models correlating air and WT using data relating to different time periods, mostly weekly and/or monthly, as the serial dependency for these timescales is generally small (e.g., Mackey and Berrie, 1991; Webb and Nobilis, 1997). That said, several studies have shown departures from linearity, showing that the rate of evaporative cooling increases at peak air temperatures, which means that the river WT will, therefore, not increase linearly with the mean air temperature (Mohseni et al.,1998, 2002), thereby demonstrating the need for more complex models and sampling of an increased number of independent variables. There are many sources of error in the modeling of river WT, including those associated with the definition of the input data and boundary conditions or with the river WT measurements used in model calibration or related to the model's structure. The predictor variables can represent a significant source of uncertainty, as river WT is not only affected by local environmental conditions, but also by upstream conditions (Moore et al., 2005). In order to minimize this source of uncertainty, some authors use a space-averaging approach in which the predictor variables consider a variety of

buffer zones with different lengths and widths (e.g., Macedo et al., 2013; Segura et al., 2014). However, the extent of the area affecting the river energy balance at a certain point is still unclear (Moore et al., 2005; Gallice et al., 2015).

In the past decades, different types of models have been successfully used to model river WT under different spatial and temporal scales. In general, the model selection depends not only on the study's requirements, namely the output timescale, but also on the availability of the input data. These include statistical models, such as the linear regression (e.g., Neumann et al., 2003; Rehana, S. and Mujumdar, P. P., 2011), multiple regression (e.g., Jeppesen and Iversen, 1987; Jourdonnais et al., 1992), non-linear regression (e.g., Mohseni et al., 1998), stochastic regression models (e.g., Ahmadi-Nedushan et al., 2007; Rabi et al., 2015) and hybrid models (statistics methods combined with physical based process, e.g., Gallice et al., 2015; Toffolon and Piccolroaz, 2015). Process-based models, based on the concepts of heat advection, transportation and equilibrium temperature are quite accurate when the boundary conditions are well characterized (e.g., Sinokrot and Stefan, 1993; Younus et al., 2000; Du et al., 2018), although they do require a large amount of forcing data, including stream geometry, air temperature, dew point temperature (or relative humidity), cloud cover and short-wave solar radiation, degree of shading and wind direction/velocity. Machine-learning (ML) models, such as artificial neural networks (ANN), have also proved to be a robust option for river WT prediction (e.g., Piotrowski et al., 2015; Temizyurek and Dadaser-Celik, 2018; Zhu et al., 2019c). In general, results show the performance of ML models to be comparable (Feigl et al., 2021; Zhu et al., 2018). Multi-layer perception neural network models are, in most cases, not outperformed by more complex and advanced neural networks models (Piotrowski et al., 2015; Zhu et al., 2019b). ML outperformed standard modeling approaches, such as multiple regression, the hybrid Air2stream model developed by Toffolon and Piccolroaz (2015) (Feigl et al., 2021), linear regression, non-linear regression and stochastic models (Zhu et al., 2018). This is not a prevailing rule, however, as the Air2stream model was also able to outperform ML, clearly indicating its potential as a valid solution in certain conditions (Zhu et al., 2019d). Table 1 describes the RMSE between observed and predicted river WT obtained from several studies and using different models. Overall, the results are quite impressive, varying from 0.42 ℃ to 2.30 ℃ in the case of the ML models. The worst results, as expected, correspond to the classical statistical models, namely multiple regression. Over/undersampling techniques are useful where regression is applicable, but the values of interest are rare or uncommon, producing an imbalanced dataset. Several available strategies exist, such as random undersampling (Torgo et al., 2015), SMOTER (Torgo et al., 2013) and Introduction of Guassian Noise (Branco et al., 2016). The SMOGN python package combines random undersampling with the two previously mentioned oversampling techniques (SMOTER and Introduction of Gaussian Noise) as a function of KNN-distances underling an observation. SMOGN was successfully implemented by Wang et al. (2021) to improve the quantification of nonlinear relationships between monthly burnt area and biophysical factors in southeast Australian forests. SMOGN was applied to resample the proportion of burnt area. This algorithm was also successfully applied by Agrawal et al. (2021) to increase a satellite imagery dataset required to identify arsenic contamination and increase the performance of ML algorithms. Although this type of solution is still not widely used, it should be considered as it has the potential to improve ML performance, particularly in cases where the forcing datasets are

small and inconsistent. From an environmental science perspective, accurate time-varying boundary conditions are vital in order to calibrate models or evaluate systems evolution. For WT calibration, this ideally means using continuous inflow temperatures, although this is complicated by the fact that WT measurements are often in short supply or completely unavailable, particularly for low-order streams. Therefore, the main objective of this study is to identify a suitable WT modeling solution for rivers with limiting forcing data. Improving this type of solution would deliver potential benefits for a wide range of environmental modeling applications, such as the analysis of seasonal/diurnal trends and biogeochemical processes in rivers based on observation datasets and the improvement of lake/reservoir water quality model boundary conditions.

It is also important to note that the studies defined to evaluate the performance of different modeling approaches are normally restricted to a very small number of test sites and usually contain a reasonable amount of forcing data (Table 1). Hence, the vital importance of increasing the number of test sites, and using a limited amount of forcing data to model river WT. The methodological approach was, therefore, defined to attempt to answer the following questions:

i)  What is the best modeling solution to predict river WT with limiting forcing data?

ii)  How does the length of the calibration period and percentage of missing data affect model performance?

iii) Can the performance of a ML model be improved through the modification of the raw training dataset with an over/undersampling technique?

iv) Is it possible to relate the modeling error with river and watershed geomorphological and hydrological variables (e.g., time of concentration; wet and dry season)?

To that end, 83 river sections with different geomorphological, meteorological and hydrological conditions were modeled. These stations correspond to all the sections for which the Portuguese Water Resources Information System (SNIRH) holds WT and discharge datasets, which are also, coincidentally, characterized by 98% missing data. The modeling ensemble includes five different models, three of which use ML algorithms optimized with a sequential model-based optimization approach: Random Forest (RF); Artificial Neural Network (ANN) and Support Vector Regression (SVR). The remaining models include the hybrid Air2stream model (using all model parametrization variations: 3, 4, 5, 7 and 8 parameters) (Toffolon and Piccolroaz, 2015) and Multiple Regression (MR). The SMOGN algorithm was also used to generate 100 synthetic samples from raw training datasets. These modified datasets were then considered to force the best model.

The results of this study will hopefully prove useful from a practical perspective by helping to improve the quality and consistency of river WT datasets.

**Table 1: List of reviewed publications on river WT modelling and the corresponding RMSE between observed and modelled WT values**

| Reference | Geographic location | Number of sites | Temporal scale | Model type | RMSE (ºC) |
|---|---|---|---|---|---|
| Chenard and Caissie, 2008 | Canada | 1 | day | ANN | 0.96 |
| DeWeber and Wagner, 2013 | Eastern U.S. | 96 | day | ANN | 1.82; 1.93 |
| Rabi et al., 2015 | Croatia | 3 | day | ANN | μ=1.70 σ=0.49; μ=2.06 σ=0.35; μ=2.30 σ=0.76 |
| Zhu et al., 2019c | U.S. | 3 | day | ANN | 0.768; 0.948; 1.242 |
| Feigl et al., 2021 | Austria; Germany and Switzerland | 10 | day | ANN | Best results: 0.45;0.42;0.43 |
| Zhu et al., 2019a | Croatia | 2 | day | ANN | 1.35; 1.70 |
| Zhu et al., 2019d | Europe, U.S. | 8 | day | ANN | [0.46,1.69] |
| Rehana, 2019 | India | 1 | day | SVR | 1.69 |
| Rajesh and Rehana, 2021 | India | 1 | day | SVR | 0.99 |
| Lu et al., 2020 | U.S. | 1 | hour | RF | 1.04 |
| Feigl et al., 2021 | Austria; Germany and Switzerland | 10 | day | RF | 0.58 |
| Rajesh and Rehana, 2021 | India | 1 | day | RF | 1.03 |
| Rehana, 2019 | India | 1 | day | MR | 1.85 |
| Moore et al., 2003 | Western Canada | 418 | year | MR | 2.1 |
| Ducharne, 2008 | France | 88 | month | MR | [1.4,1.9] |
| Zhu et al., 2019a | Croatia | 2 | day | MR | 2.33; 2.74 |
| Toffolon and Piccolroaz, 2015 | Switzerland | 3 | day | Air2stream | 3-par[0.88, 1.05];4-par[0.87,1.04] 5par[0.70, 1.05]; 7-par[0.65,0.78];8par[0.75,0.62] * |
| Zhu et al., 2019d | Europe, U.S. | 8 | day | Air2stream | 3-par[0.64, 1.25]; 5-par[1.31, 0.76]; 8par[1.37;0.93] * |
| Feigl et al., 2021 | Austria; Germany and Switzerland | 10 | day | Air2stream | 8-par[0.74,1.17] * |

*The model can be applied with 3, 4, 5, 7 or 8 parameters (3-par; 4-par; 5-par; 7-par and 8-par)

## 2 Study area and data

The watersheds considered in this study are located in Portugal (Fig. 1). This southern European country has a typical Mediterranean climate. Maximum daily mean air temperature ranges from 13 ºC in the central highlands to 25 ºC in the southeastern region. The minimum daily mean air temperature ranges from 5 ºC in the northern and central regions to 18 ºC in the south (Soares et al., 2012). The spatial and temporal heterogeneity of precipitation, which differs from a relatively wet annual maximum of over 2 500 mm/yr, in the mountainous northwest to a much drier 400 mm/yr. in the flat southeast, is defined by complex topography and coastal processes (Cardoso et al., 2013; Soares et al., 2015).

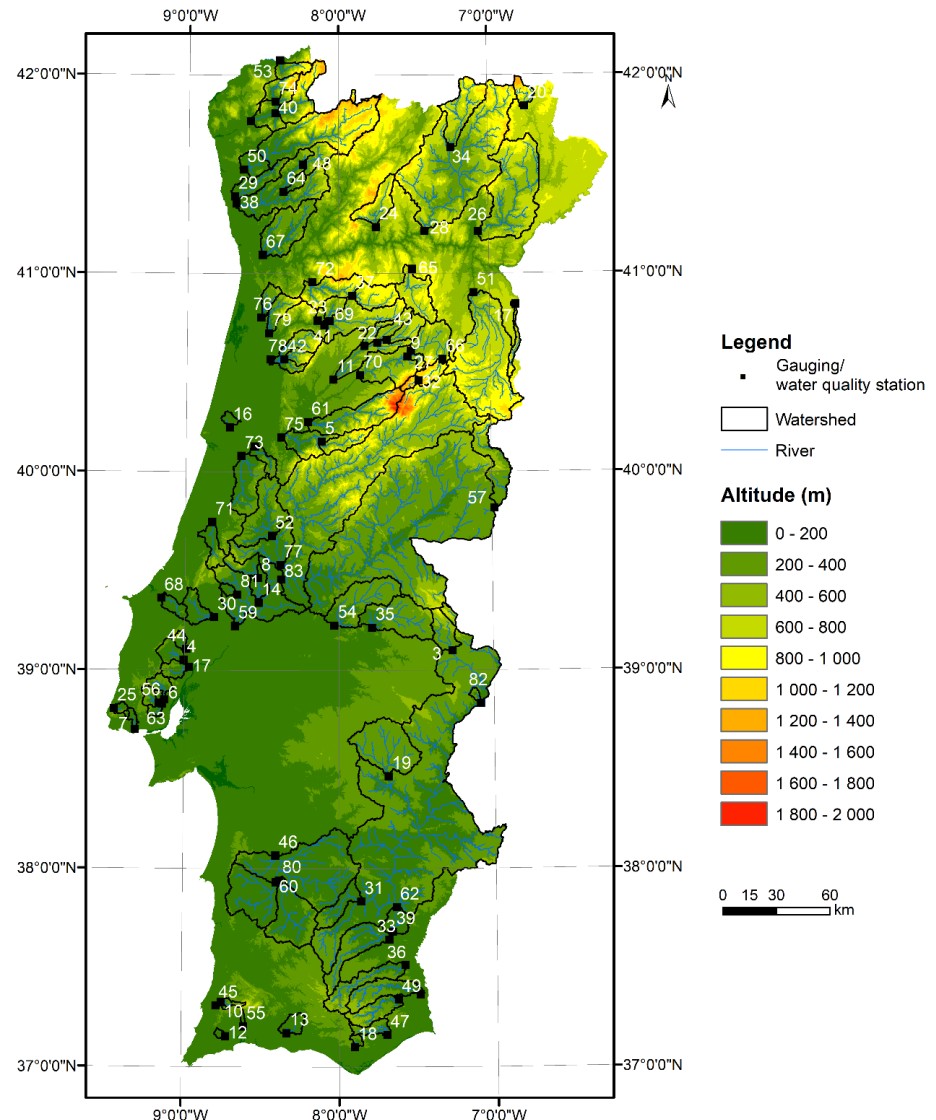

**Figure 1: Location of the watersheds considered in the study (from DEM – Shuttle Radar Topography Mission (Farr et al., 2007))**

The models used in this study were forced with daily mean, maximum and minimum air temperature and global radiation values obtained from the fifth-generation atmospheric reanalysis, ERA5-Land, produced by the European Centre for Medium-Range Weather Forecasts (ECMWF). ERA5-Land is the ECMWF's most advanced reanalysis dataset for land applications (Muñoz-Sabater, 2019; Muñoz-Sabater, 2021). The horizontal resolution of this dataset (0.1° x 0.1°; Native resolution is 9km.) is higher than that corresponding to ERA-Interim and ERA5 (0.28° x 0.28°; Native resolution 31km grid). The vertical coverage ranges from 2m above surface level to a soil depth of 289cm. The Carbon Hydrology-Tiled

ECMWF Scheme for Surface Exchanges Over Land (CHTESSEL) forced with atmospheric forcing derived from ERA5
near-surface meteorology state and flux fields (10m above ground level) is central to ERA5-Land. The surface fluxes are
linearly interpolated from the ERA5 resolution of approximately 31km to the ERA5-Land resolution of 9km. Land
characteristics, such as soil and vegetation type and vegetation cover are described by time-invariant fields (Muñoz-Sabater
et al., 2021a). The air temperature reanalysis dataset (hourly data) covering a period of 42 years (1 January 1980 to 31
December 2021), was downloaded from the Copernicus Climate Change Service (C3S) Climate Data Store (Muñoz-Sabater,
2019; Muñoz-Sabater, 2021). The watershed discharge data used to force the models and the WT considered for the model's
validation are also available from SNIRH (http://snirh.apambiente.pt). The SNIRH provides data and WT values for 2471
water-quality stations, of which only 98 have gauging stations with discharge values, one of the conditions required to
implement the Air2stream model. The missing discharge data was replaced with the corresponding climatological year
value, hence only the gauging stations with data spanning at least a full year (365/366 values) were kept. Following this
initial analysis, the number of stations considered was reduced to 83. Data availability varies from station to station but
generally covers a period of 42 years (1980-2021). However, a significant amount of daily river WT values are missing,
ranging from 96.9% to 99.9% ($\mu$= 98.8%; $\sigma$=0.68).

Table 2 shows the number of WT for the annual data series, for the dry season (April to September) and for the wet season
(October to March) separated into training and test datasets, considering all stations

**Table 2: Number of WT for the annual, dry- and wet season, training and test data series**

| Temporal scale | Phase | Total number | Mean | Standard deviation | Maximum | Minimum |
|---|---|---|---|---|---|---|
| Annual | Train | 8384 | 101 | 60 | 237 | 11 |
| Annual | Test | 3593 | 43 | 26 | 102 | 5 |
| Dry season | Train | 4161 | 50 | 32 | 116 | 4 |
| Dry season | Test | 1783 | 21 | 14 | 50 | 2 |
| Wet season | Train | 4223 | 51 | 29 | 124 | 4 |
| Wet season | Test | 1810 | 22 | 13 | 53 | 2 |

## 3 Methodology

The definition of the methodological approach was supported by the following:

i) It is important to model a significant number of watersheds to reduce the degree of results uncertainty. This was minimized by modeling all the watersheds located in Portugal for which river WT and discharge values were available;

ii) The number and type of models is also key to gaining a comprehensive understanding of the structural differences between the models and their performance. The five models considered in this study include state-of the art algorithms, with one classic modeling approach (MR) included to establish a benchmark;

iii) Generally-speaking, there are no available sources of observed meteorological data for either the watershed or the area surrounding the lowest part of low-order rivers and, as such, the forcing meteorological datasets considered in this study were obtained from the ERA5-Land reanalysis.

The modeling reference is the watershed main gauging station/water-quality station. Therefore, the hourly air temperature (°C) and global radiation (shortwave) ($Jm^{-2}$) input datasets of the nearest ERA5-Land grid point were initially downloaded before the air temperature datasets were corrected according to the gauging station and the ERA5-Land grid point altitude. This correction was achieved by considering a linear variation of air temperature with the altitude, $\frac{dT}{dz} = -6.0, °C/km$ (Fahrer and Harris, 2004). After this correction, the mean, maximum and minimum daily air temperature values and the mean global radiation values were computed from the hourly meteorological datasets. Initially the model predictors were selected on the basis of their availability and the results obtained with other studies (e.g., Zhu et al., 2019c; Feigl et al., 2021). These included, mean, max. and min. daily air temperature (°C), mean daily total radiation (shortwave) ($Jm^{-2}$), discharge ($m^3.s^{-1}$) and two temporal features, the month (0-12) and the day (1-365) of the year (MOY and DOY, respectively) (Table 3).

**Table 3: Model predictor variables**

| Model | Predictor variables | Output variable |
|---|---|---|
| RF | Mean, max., and min. daily air temperature (ºC)<br>Mean daily total radiation (shortwave) (Jm$^{-2}$)<br>Mean daily discharge (m$^3$.s$^{-1}$)<br>MOY and DOY | |
| ANN | Mean, max., and min. daily air temperature (ºC)<br>Mean daily total radiation (shortwave) (Jm$^{-2}$)<br>Mean daily discharge (m$^3$.s$^{-1}$)<br>MOY and DOY | |
| SVR | Mean, max., and min. daily air temperature (ºC)<br>Mean daily total radiation (shortwave) (Jm$^{-2}$)<br>Mean daily discharge (m$^3$.s$^{-1}$)<br>MOY and DOY | Water temperature |
| Air2stream | Mean daily air temperature (ºC)<br>Mean daily discharge (m$^3$.s$^{-1}$) | |
| MR | Mean, max., and min. daily air temperature (ºC)<br>Mean daily total radiation (shortwave) (Jm$^{-2}$)<br>Discharge (m$^3$.s$^{-1}$)<br>MOY and DOY | |

The results section starts with the evaluation of the ERA5-Land mean daily air temperature datasets. These datasets were compared with ground measurements of mean daily air temperature considering all the meteorological datasets located within a 5km radius of the stations considered in this study. Following this initial analysis, the models (*vide* Sect. 3.1 to 3.6) were applied to each of the 83 input datasets, divided between a training (70% of the entire dataset) and testing dataset (the remaining 30%). The validation phase was not considered due to size of the available datasets. It should be noted that, in

the case of the Air2stream model, 70% of the initial dataset corresponds to the calibration dataset and the remaining 30% to the validation dataset. Hyperparameter optimization was achieved for the ML models through the application of the Tree-structured Parzen Estimators (TPE) algorithm (vide Sect. 3.6). Given the large number of input datasets and the fact that the optimization process can be very time consuming, the following approach was implemented (Fig. 2):

i) The 83 stations were ordered as a function of the number of samples (lowest to highest) and were divided into four

different classes (L ≤ 50; 50< L ≤100; 100< L ≤200; L>200). Three stations were selected within each class: 1) the station with the least samples; 2) the station with the most samples; 3) the station with the number of samples that most closely corresponded to the average sample number for each class. The 12 datasets selected corresponded to Stations 1, 7, 12, 13, 22, 29, 30, 46, 59, 60, 73 and 83;

ii) The ML and TPE algorithms were applied to the 12 datasets. At this stage there were 12 optimized model structures

computed with the TPE algorithm for each ML model;

iii) The 12 optimized models obtained for each ML were subsequently applied to the 83 datasets and the best performing model at each station was calculated on the basis of the computed root mean square value (RMSE). Hence, the ensemble of the best results obtained across the 12 different models for the 83 stations defines the overall ML results.

To evaluate the possibility of further improving the results obtained with the best model, 100 different training datasets were then derived for each of the 12 datasets selected in step i) through the application of the Synthetic Minority Over-Sampling Technique for regression with Gaussian Noise (SMOGN) (Branco et al., 2017) (Section 3.7). The five SMOGN parameters that drive the algorithm were randomly derived within each model run considering a pre-defined parameter space (Table A2). A description of the model parameters is included in Table A2. The best ML model obtained in step iii) was then forced with the modified training datasets (100 for each station) and optimized with TPE.

Following this analysis, and in order to further investigate the relevance of the predictor variables, the input feature importance was estimated for all stations by considering the best performing model. Additionally, the best model was used to evaluate the differences between observed and model river WT considering the sequential increase of the models' predictors: 1) mean air temperature 2) mean air temperature + discharge; 3) mean air temperature + discharge + radiation; 4) mean air temperature + discharge + radiation + maximum air temperature; 5) mean air temperature + discharge + radiation + maximum air temperature + minimum air temperature; 6) mean air temperature + discharge + radiation + maximum air temperature + minimum air temperature + MOY; 7) mean air temperature + discharge + radiation + maximum air temperature + minimum air temperature + MOY + DOY.

The effect of the watershed geomorphological and hydrological variables was addressed with the analysis of the watershed time of concentration, a variable that encapsulates some of the main watershed characteristics that effect the river WT. The well-known Temez equation (Temez, J.R., 1978) (Sect. 3.8) initially defined for small-scale Mediterranean watersheds was selected for this analysis. Additionally, the Gaussian Mixture Models algorithm implemented with the machine-learning python package, scikit-learn (Pedregosa et al., 2011), was used for cluster analysis. The algorithm assumes that the data points belong to a mixture of Normal distributions. The covariance structure of the data as well as the center of the distributions are used to compute probabilistic cluster assignments.

The results from the various models were evaluated with six metrics considering the observed and predicted daily datasets of river WT. During the results evaluation three types of datasets were considered:

Annual datasets: All available daily averages of WT are compared to field data;

Wet season: Only the daily averages of WT corresponding to the wet season are compared to field data (October to March);

Dry season: Only the daily averages of WT corresponding to the dry season are compared to field data (April to September).

The metrics were selected in order to provide not only a consistent interpretation of the models' results, but also to facilitate comparison with the results obtained in other studies (Sect. 3.9). The following sections describe each of the models and outline their relevant advantages/disadvantages.

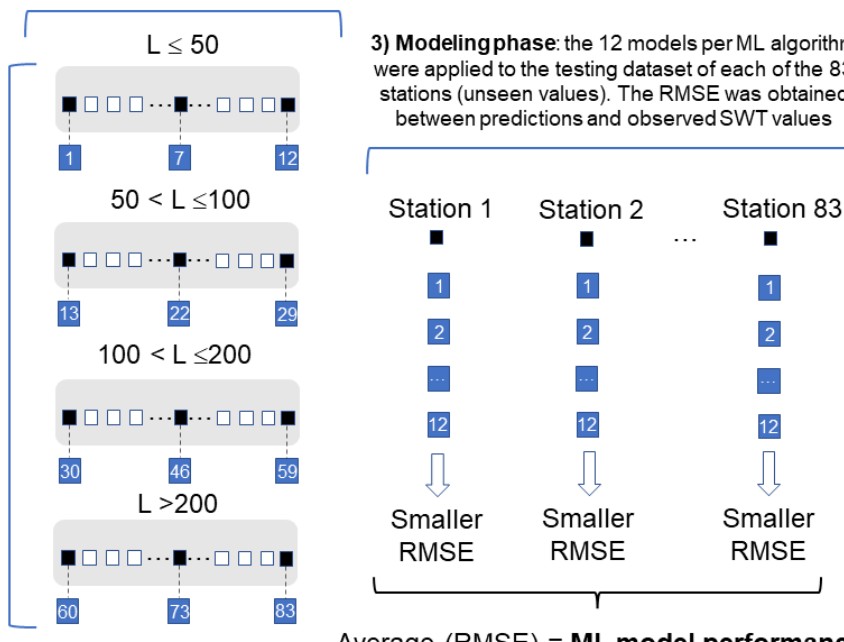

**2) Model definition phase**: 12 models per ML algorithm were optimized with Hyperopt for 12 datasets

**3) Modeling phase**: the 12 models per ML algorithm were applied to the testing dataset of each of the 83 stations (unseen values). The RMSE was obtained between predictions and observed SWT values

**1) Stations selection:**
Stations were ordered as a function of the number of samples (lowest to highest) and 4 classes were defined. Within each class 3 stations were selected:
**i)** with the least samples;
**ii)** with the most samples;
**iii)** with the number of samples that most closely corresponds to the average sample number for each class.

Average (RMSE) = **ML model performance**

**Figure 2: Schematic and simplified representation of the modeling process. Initially, 12 stations were selected as a function of the number of samples they contained. The ML models were trained and optimized for the 12 station datasets (Model definition phase). The ML models were then applied to the 83 stations (Modeling phase). The ensemble of the best results as a function of the RMSE describes the final ML results per station.**

### 3.1 Random Forest

The RF algorithm (Random Forest Regressor) was implemented with the machine learning python package, scikit-learn (Pedregosa et al., 2011). This model fits classifying decision trees on various sub-samples of the datasets and then combines the predictions. Decision trees can model complex non-linear relations. The algorithm uses averaging to control overfitting

and improve the algorithm predictive accuracy, thus effectively balancing the bias- variance trade-off. They are robust to outliers, missing values and irrelevant or noisy variables because the model implicitly performs feature selection and generates uncorrelated decision trees. Beyond these advantages, there is one major drawback, common to all the ML methods, with results difficult to interpret due to the intrinsically black box nature of the algorithm. More details about RF can be found in the literature (Breiman, 2001, Louppe, 2014).

### 3.2 Artificial Neural Network

The ANN prototyping and building was achieved with the NeuPy python library (Shevchuk, Y., 2015). This library uses Tensorflow (an open-source platform for machine learning) as a computational backend for deep learning models (Abadi et al., 2015). The momentum algorithm was selected for the ANN implementation because of the improved control it provides with regard to overfitting. This is an iterative first-order optimization method that uses gradient calculated from the average loss of a neural network. This algorithm promotes a gradual transition in the balance between stability and rate of change (Qian, 1999), the result is faster convergence and reduced oscillation. ANN has been successfully used to model river WT (Chenard and Caissie, 2008; DeWeber and Wagner, 2014; Piotrowski, et al., 2015). This type of model is reasonably accurate and does not require a large number of input variables but does have two significant drawbacks. The model has no capacity to provide information on energy flux mechanisms within the river and has a tendency to overfit the training dataset, thereby considerably diminishing the model's ability to generalize the features or patterns present in the training dataset (Srivastava et al., 2014). For the implementation of the model, the training data was shuffled before training and the weights were randomly initiated. The loss function included the MSE to measure the accuracy of the results, as well as L2 regularization and dropout layers to minimize overfitting. The step decay algorithm was used to regularize the learning rate.

### 3.3 Support Vector Regression

The Epsilon-Support Vector Regression algorithm was also implemented using the machine-learning python package, scikit-learn (Pedregosa et al., 2011). This type of algorithm is generally characterized by the use of kernels functions, sparseness of the solution and the absence of a local minimum (Platt, 1998; Smola et al., 2004). The algorithm searches for a line or hyperplane in multidimensional space that divides two or more variables. The hyperplane with the optimum number of points is the best fit (Awad and Khanna, 2015). The SVR training relays on the use of a symmetrical loss function, which penalizes high and low errors. The algorithm also ignores errors that are less than a certain threshold, $\varepsilon$. According to Awad and Khanna (2015), the computational complexity of the algorithm does not depend on the dimensionality of the input space, which is a relevant advantage. It also offers good prediction accuracy and excellent generalization capability. Regardless of the advantages, this algorithm can be computationally expensive, which can be a significant drawback.

### 3.4 Air2stream

The Air2stream model solves a lumped heat-exchange budget between an unknown river section volume, its tributaries, groundwater, and the atmosphere (Toffolon and Piccolroaz, 2015). The river WT variation is described by the following equation:

$$\rho C_p V \frac{dT_W}{dt} = AH + \rho C_p \left( \sum_i Q_i T_{W,i} - Q T_w \right), \tag{1}$$

Where $T_w$ is the WT of a river section with a volume V and surface area A, ρ and $C_p$ are the water density and the specific heat capacity, respectively. H is the net heat flux at the air-water interface, and $T_{Wi}$ is the i-th WT of the discharge $Q_i$ tributary or groundwater. The model assumes that air temperature can be used as a proxy for all surface heat fluxes. A Taylor series expansion is used to include the overall effect of air temperature. Q is the discharge downstream of the river section and t is time. Eq. (2) is the simplified form of Eq.(1) (Toffolon and Piccolroaz, 2015). This equation, with 8 parameters, forms the basis of the Air2stream model:

$$\frac{dT_W}{dt} = \frac{1}{\theta^{a_4}}\left(a_1 + a_2\,T_a - a_3 T_w + \theta\left(a_5 + a_6 cos\left(2\pi\left(\frac{t}{t_y} - a_7\right)\right) - a_8 T_w\right)\right),\tag{2}$$

Where $Ta$ is the air temperature, θ is the dimensionless discharge ($\theta = Q/\overline{Q}$ (3), $\overline{Q}$ is the mean discharge. The parameter, $a_4$ is related with the exponent of the rating curve. The model is fitted to the entire input dataset (air temperature, WT and discharge) and the value of $a_4$ and the value of all others model parameters are estimated during the model optimization process (calibration phase). In this study the Crank Nicolson scheme was used to solve the model equation. Following, Toffolon and Piccolroaz, 2015, the model parameters were estimated using the Particle Swarm Optimization method with inertia weight (Shi and Eberhart, 1998) with a population size of 2000 particles and 2000 iterations. In this study five versions of this model were considered to model WT. The 3, 4, 5, 7 and 8 parameter versions. Please refer to Toffolon and Piccolroaz (2015) for a full description of each one of the models' parameterizations.

## 3.5 Multiple regression

This model was implemented using the machine learning python package, scikit-learn (Pedregosa et al., 2011). In this model the predicted value is expected to result in a linear combination of the input features:

$$\hat{y} = w_0 + w_1 x_1 + w_2 x_2 + \cdots + w_p x_p,\tag{4}$$

Where, $\hat{y}$ is the predicted value, $w_0$ is $\hat{y}$-intercept (constant term), $w_1$ to $w_p$ are the model coefficients, and $x_1$ to $x_p$ are the model input features. The model fits a linear model with coefficients $w_1$ to $w_p$ to minimize the residual sum of squares between the observed and predicted values.

## 3.6 Hyperparameter optimization

Hyperparameter optimization was achieved using the Tree-structured Parzen Estimators (TPE) algorithm implemented with the Hyperopt library (Bergstra et al., 2013). The optimization process is initiated with the selection of a prior distribution (e.g., uniformly distributed), then, for the first iterations, the TPE algorithm is warmed up with some random iterations (Random Search). After this initial set up the algorithm collects new observations and on completion of the iterations it selects the set of parameters that it will try during the next iteration. The algorithm scores and divides the collected

observations into two groups. The first group includes the best observations and the second group all the others. The main
objective is to identify a set of parameters most likely to be in the first group. The TPE algorithm can serve as a good alternative to the Gaussian Process (GP) with expected improvement (EI) as it fixes some of the disadvantages associated with the latter. It can be difficult to select the right hyperparameters for GP with EI due to the many different Kernel types associated with this process. TPE uses simpler Kernels as a building block, which facilitates hyperparameter selection. Furthermore, TPE is faster than GP with EI when the number of hyperparameters increases. One notable drawback, however, is that the TPE algorithm selects parameters independently from each other. It is a well-known fact that the number of epochs of an ANN and regularization are related and that these two parameters influence the overfitting to a significant degree. To overcome this problem two different choices for the epochs, with and without regularization, were constructed. TPE hyperparameter optimization consists of 20 random parameter samples and 200 iterations. The hyperot algorithm samples 1000 candidates and selects the candidate that has the highest expected improvement (n_EI_candidates = 1000). The coefficient of determination ($R^2$) was considered as the algorithm score. The algorithm uses 20% of best observations to estimate the next set of parameters (gama = 0.2). Table A1 shows the models parameters and the corresponding optimization range.

## 3.7 Synthetic Minority Oversampling Technique for Regression with Gaussian Noise (SMOGN)

SMOGN (Branco et al., 2017) is highly effective when working with imbalanced regression datasets. The algorithm applied with the Python implementation obtained from the SMOGN GitHub repository (SMOGN, 2022) combines random undersampling with two oversampling approaches: The Synthetic Minority Oversampling Technique for Regression (SMOTER) (Torgo et al., 2013) and the Gaussian Noise (SMOTER-GN) techniques (Branco et al., 2016). The algorithm selects between two sampling techniques, considering the K-Nearest Neighbors (KNN) distances underlying an observation: if the distance is too great, SMOTER-GN is applied, otherwise SMOTER is applied. By combining both approaches for generating synthetic samples the authors made the decision to apply SMOGN; a more conservative approach which would minimize the potential risks incurred with SMOTER (Branco et al., 2017). Table A2 describes the parameter search space considered to derive the model datasets.

### 3.8 Time of concentration
The time of concentration was estimated using the Temez equation (Temez, J.R., 1978), which was defined for small natural watersheds located in Spain. In this equation, $T_C$ is the time of concentration in hours, L is the length of the main water line, in km, and $J$ is the mean steepness (ratio between the mean fall and the L length of the water line), m/m.

$$T_C = 0.3 \left(\frac{L}{J^{1/4}}\right)^{0.76},$$
(5)

### 3.9 Evaluation metrics

Model assessment was performed with six different metrics: the mean absolute error (MAE), the root mean square error (RMSE), the Nash-Sutcliffe efficiency (NSE) (Nash and Sutcliffe, 1970), the Kling-Gupta efficiency (KGE) (Kling et al., 2012), the bias (BIAS) and the coefficient of determination ($R^2$). The metrics were computed using the following equations, where $m_i$ and $o_i$ are the modeled and observed values, $\bar{m}$ and $\bar{o}$ their means, $\sigma_m$ is the standard deviation of the modeled values, $\sigma_o$ the standard deviation of the observed values and r is the Pearson coefficient:

$$MAE = \frac{1}{N}\sum_{i=1}^{N}|m_i - o_i|, \tag{6}$$

$$RMSE = \sqrt{\frac{1}{N}\sum_{i=1}^{N}(m_i - o_i)^2}, \tag{7}$$

$$NSE = 1 - \left[\frac{\sum_{i=1}^{N}(o_i - m_i)^2}{\sum_{i=1}^{N}(o_i - \bar{o})^2}\right], \tag{8}$$

$$KGE = 1 - \sqrt{(r-1)^2 + \left(\frac{\sigma_m}{\sigma_o} - 1\right)^2 + \left(\frac{\bar{m}}{\bar{o}} - 1\right)^2}, \tag{9}$$

$$Bias = \bar{m} - \bar{o}, \tag{10}$$

$$R^2 = \frac{\sum_{i=1}^{N}(m_i - \bar{o})^2}{\sum_{i=1}^{N}(o_i - \bar{o})^2} \times 100, \tag{11}$$

The RF and ANN algorithms use the mean square error to measure the results accuracy:

$$MSE = \frac{1}{N}\sum_{i=1}^{N}(m_i - o_i)^2, \tag{12}$$

### 4 Results

#### 4.1 Air temperature - ERA5-Land *versus* ground observed datasets

In this analysis the observed air temperature datasets of a total of eleven meteorological stations were considered. These are all the available air temperature datasets observed within a 5km radius of the stations considered in this study. The results show that the mean RMSE obtained between the two datasets considering all stations varied from 1.26ºC to 2.05ºC (μ=1.54ºC; σ=0.24ºC) and that, according to the mean bias values, the ERA-Land datasets tend to overestimate the observed air temperature datasets at 91% of the stations. Overall, a mean RMSE value of 1.54ºC (σ=0.24ºC) and a mean NSE value

of 0.90 (σ=0.07) is indicative of a good performance. This conclusion corresponds to the results obtained in other studies, namely, Vannela et al. (2022) (Italy/3 regions – RMSE: 1.76ºC, 1.82ºC and 1.97ºC); Araújo et al. (2022) (Brazil/3 regions – RMSE: 0.60ºC, 1.11ºC and 0.41ºC); Zhao et al. (2022) (China/1 region –2.2ºC). However, as shown in Figure 3, several significant sporadic discrepancies were produced between the two datasets. The results also show a nationwide distribution of stations with an RMSE of over 2ºC. Generally, these results suggest that the consideration of the ERA5-Land air temperature datasets for WT modeling can, sporadically, induce some significant discrepancies between the two datasets.

**Table 4: Evaluation of ERA5-Land daily air temperature datasets - MAE, RMSE, NSE, KGE, bias and R$^2$ (with standard deviation) between observed and ERA5-Land values**

| Station | Number of dataset values | MAE, ºC | RMSE, ºC | NSE | KGE | Bias, ºC | R$^2$ |
|---------|--------------------------|---------|----------|-----|-----|----------|-------|
| st4 | 80 | 1.10±0.26 | 1.39±0.28 | 0.91±0.04 | 0.94±0.05 | 0.74±0.47 | 0.94±0.02 |
| st6 | 120 | 1.10±0.37 | 1.34±0.38 | 0.90±0.17 | 0.92±0.09 | -0.15±0.79 | 0.90±0.11 |
| st30 | 98 | 1.31±0.29 | 1.72±0.40 | 0.91±0.07 | 0.95±0.06 | -0.48±0.70 | 0.92±0.05 |
| st32 | 67 | 1.16±0.52 | 1.43±0.58 | 0.96±0.04 | 0.94±0.05 | -0.75±0.90 | 0.97±0.02 |
| st38 | 110 | 0.88±0.34 | 1.26±0.48 | 0.94±0.09 | 0.96±0.06 | -0.46±0.57 | 0.95±0.04 |
| st42 | 21 | 1.19±0.47 | 1.53±0.58 | 0.93±45.49 | 0.87±2.22 | -0.42±0.75 | 0.94±0.03 |
| st50 | 90 | 1.08±0.30 | 1.45±0.48 | 0.91±0.06 | 0.89±0.11 | -0.14±0.39 | 0.92±0.04 |
| st62 | 24 | 1.30±0.74 | 1.67±0.80 | 0.89±6.28 | 0.94±0.92 | -0.17±1.36 | 0.90±0.04 |
| st68 | 47 | 1.60±1.18 | 2.05±1.09 | 0.71±9.81 | 0.86±0.23 | -1.47±1.24 | 0.88±0.2 |
| st83 | 137 | 1.49±0.40 | 1.79±0.39 | 0.92±0.03 | 0.94±0.04 | -0.60±0.88 | 0.93±0.02 |
| st91 | 51 | 1.04±0.13 | 1.33±0.16 | 0.93±0.04 | 0.96±0.09 | -0.46±0.47 | 0.94±0.04 |

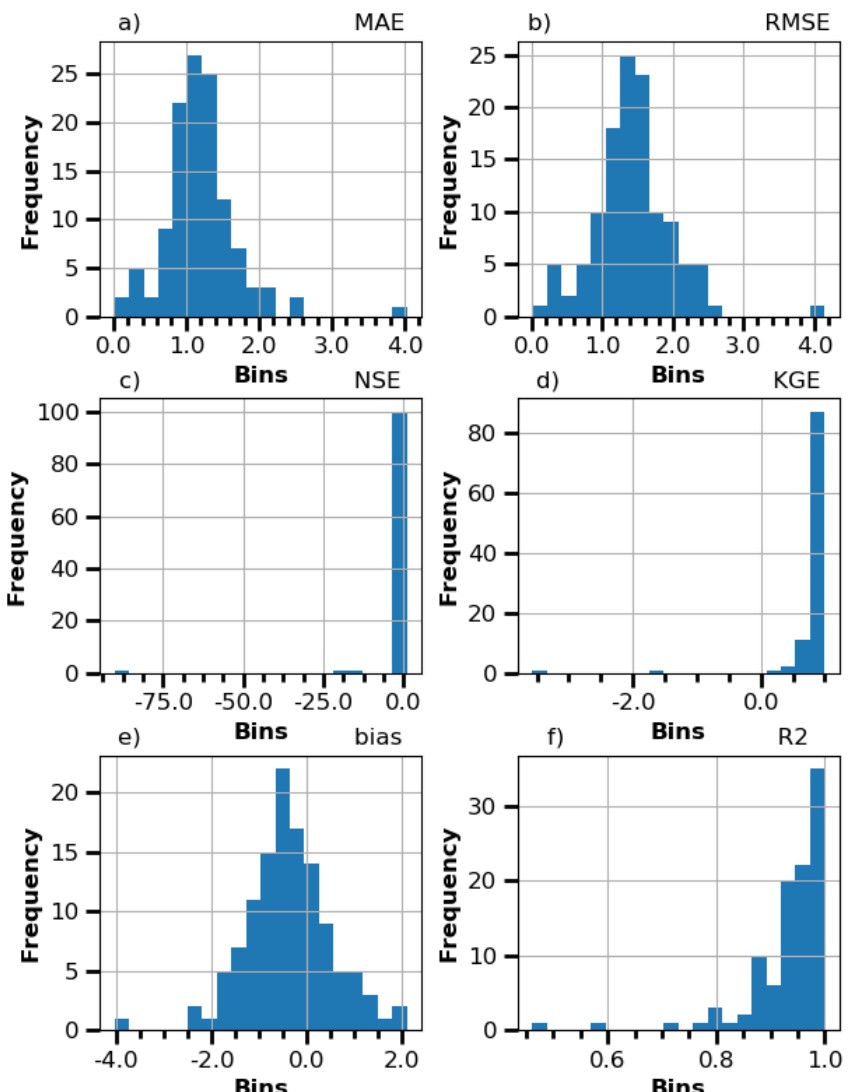

**Figure 3: Metrics histograms of daily air temperature - ERA5-Land *versus* ground observed datasets**

**4.2 Model intercomparison – annual datasets**

The results obtained from all the models for the testing phase and the annual datasets showed the RF model ensemble, with a mean RMSE of 3.18 ºC (σ = 1.06) to be the top performing model, with the ANN model ensemble, with a mean RMSE of 3.22 ºC (σ = 1.05), and the SVR model ensemble, with a mean RMSE of 3.37 ºC (σ = 0.96), ranked second and third, respectively (Table A3). The SVR model produced the lowest RMSE of all the simulations run: 1.34 ºC for Station 8 with a training dataset of 20 values (SVR parameters: kernel = 'sigmoid', degree = 3, C= 1000, gamma=0.0001, epsilon=0.005). The RF was also the best performing model based on a single model run (RF parameters: n_estimators = 50, max_depth = 485, min_samples_split = 5, max_features = 'auto', bootstrap = True), with a mean RMSE of 3.37 ºC (σ = 0.96).

The Air2stream model with 3-parameters is the best of the hybrid model parameterizations, with a mean RMSE of 4.06 ºC (σ = 1.17), followed by the MR, with an annual mean RMSE of 4.28 ºC (σ = 1.17). The NSE, KGE and $R^2$ values are closely aligned with the RMSE variation among the different models. Considering the performing ratings defined by Moriasi et al., (2007), the results obtained with the RF model ensemble, as described by the mean annual NSE value (μ=0.52; σ=0.23) can be considered satisfactory (0.50<NSE<0.65). According to the same classification, the ANN and the SVR with a mean annual NSE value of 0.48 (σ=0.28) and 0.47 (σ=0.19) produce an unsatisfactory modeling performance (NSE ≤ 0.50). The same classification was obtained with the all the parameterizations of the Air2stream model and the MR, but with a significantly reduced NSE value. The mean annual RMSE for the ensemble of all model results for the testing phase was 2.75 ºC (σ = 1.00), varying from 1.34 ºC to 6.03 ºC. Therefore, according to the mean NSE value (μ=0.56; σ=0.48), the model ensemble can be considered satisfactory. The contribution of the individual models to the results ensemble considering the stations with the lowest mean annual RMSE was as follows: RF: 35; ANN: 17; SVR: 14; Air2stream (3 par):1; Air2stream (8 par):2; MR:14. It is important to mention that these results are not correlated with the number of values in the training or testing datasets but are a consequence of the dataset's quality and of the model's performance.

Figures 4 and 5 show the RMSE obtained with each model during the training and testing phases, respectively. The interannual variability is described by the standard deviation. The stations are ordered as a function of the number of training and testing datasets, from the smallest to the largest.

The results help to explain the performance of the models during the testing phase by showing that:

i)   During the training phase, all models exhibited a very low mean RMSE and interannual variability, except the Air2stream (3par) and the MR;

ii)  The RF underfitted the training datasets with less than 30 values and, consequently, the predicted WT values exhibited a high RMSE and interannual variability during the testing phase (σ=1.28) (Fig. 4 and 5);

iii) During the training phase, the ANN exhibited the lowest mean annual RMSE (μ=0.44ºC; σ=0.40) (Table 4). This model clearly overfitted the training datasets, with less than 30 values, which increased the RMSE obtained for Stations 1 to

11 (Fig. 4 and 5). The model mean RMSE variability during the testing phase is equal to that obtained for the RF, which exhibited the lowest variability during the testing phase ($\sigma=1.28$);

iv) Like the ANN, the SVR overfitted the training datasets of the first 10 stations although the model had the lowest mean RMSE interannual variability during the testing phase ($\sigma=1.25$), including for the stations 1 to 10;

v) The Air2stream (3-parameters) model and the MR exhibited the highest mean RMSE and interannual variability during both phases. In fact, the MR exhibited a significant degree of interannual variability ($\sigma=4.10$) for the datasets with less than 30 values (Stations 1 to 10), which was reflected in the results obtained during the testing phase.

Fig. 6 was included to provide greater insight into the underfitting and overfitting associated with the ML models. The training datasets with less than 30 values are clearly underfitted by the RF model (Fig. 6a) and overfitted by the ANN and SVR (Fig. 6c and 6e). In the case of the ANN and the SVR, the overfitting is stronger and more closely correlated with the number of training datasets (RF: $R^2 = 0.13$; ANN: $R^2 = 0.52$; SVR: $R^2 = 0.58$).

It is also interesting to look at the results obtained from the models with regard to levels of performance. Fig. 7 shows the temporal evolution of the WT values obtained during the training and testing datasets for Station 59 (138 training values) and 2 (11 training values). Based on the RF model results, these are the stations with the best and worst mean annual RMSE. There are clear, fundamental differences between the ML models and the Air2stream and MR models. The ML models are highly effective. They describe a large number of spurious observed values in the WT values that can be associated with the sub-daily variation of the river WT, underground inflows or with a monitoring error and, by doing so, the predicted temporal evolution of the river WT oscillates widely (Fig. 7a, 7c, and 7e). This was not the case with the Air2stream or MR models. The results obtained from these two models demonstrate the fact that, in the absence of quality input training information (quantity plus quality), their predictive performance is significantly lower than that of the ML models. This is illustrated by the less oscillating sinusoidal evolution of the river WT (Fig. 7g and 7i). When considering very small training datasets, such as the dataset corresponding to Station 2, with 11 training values and 5 testing values, ML models tend to have a very unrealistic response as they either overfit or underfit the training datasets (Fig. 7b, 7d and 7f). In this example, the Air2stream (5-parameters) model has a delayed but more realistic response. The MR performed the worst, with the model unable to describe the correlation between the predictor variables and the observed river WT (Fig. 7j).

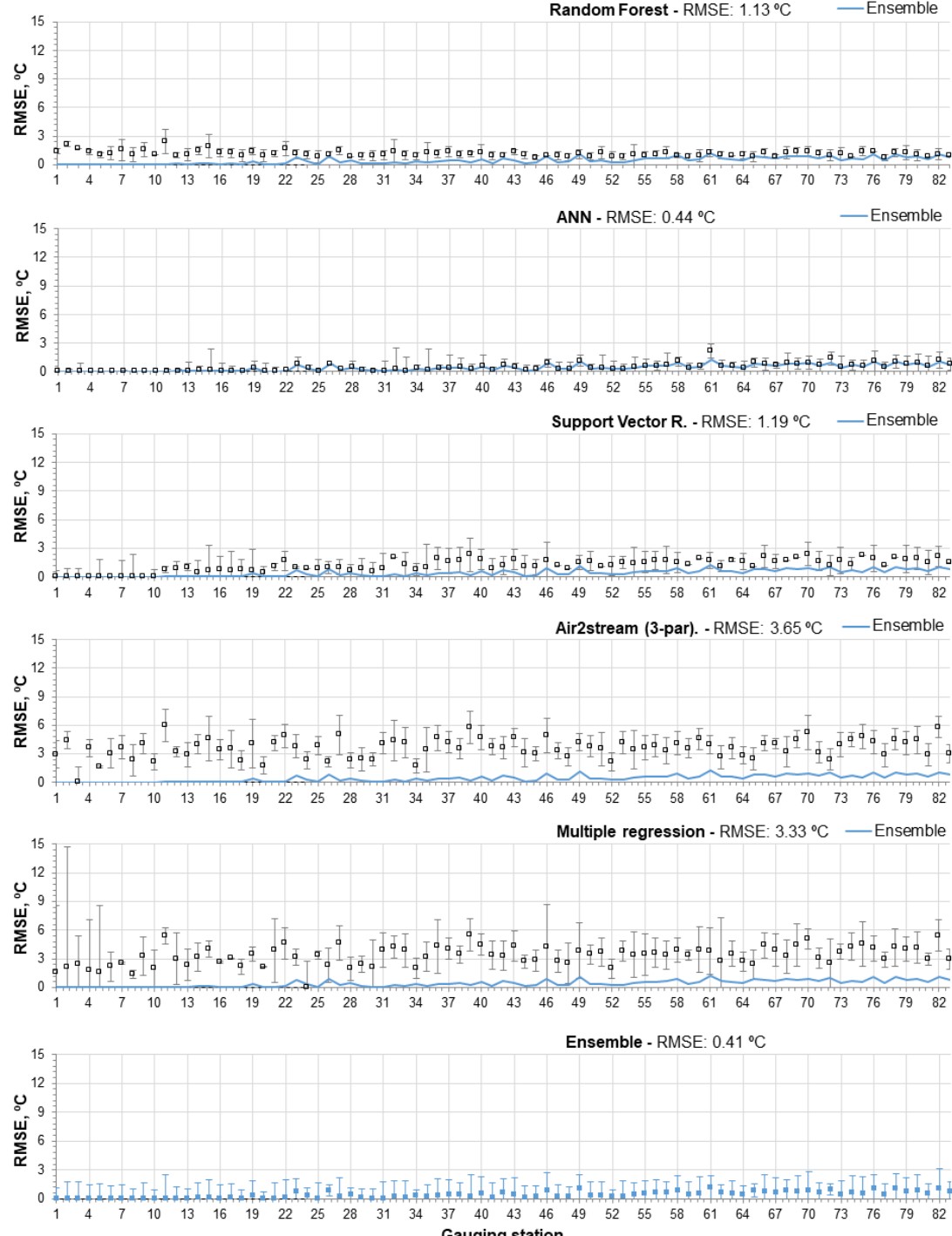

Figure 4: Root-mean-square error between observed and predicted WT values obtained during the training phase with all models (with standard deviation of interannual RMSE), considering the model results and the ensemble of all models results. Stations are ordered by the number of training dataset values, from smallest to largest

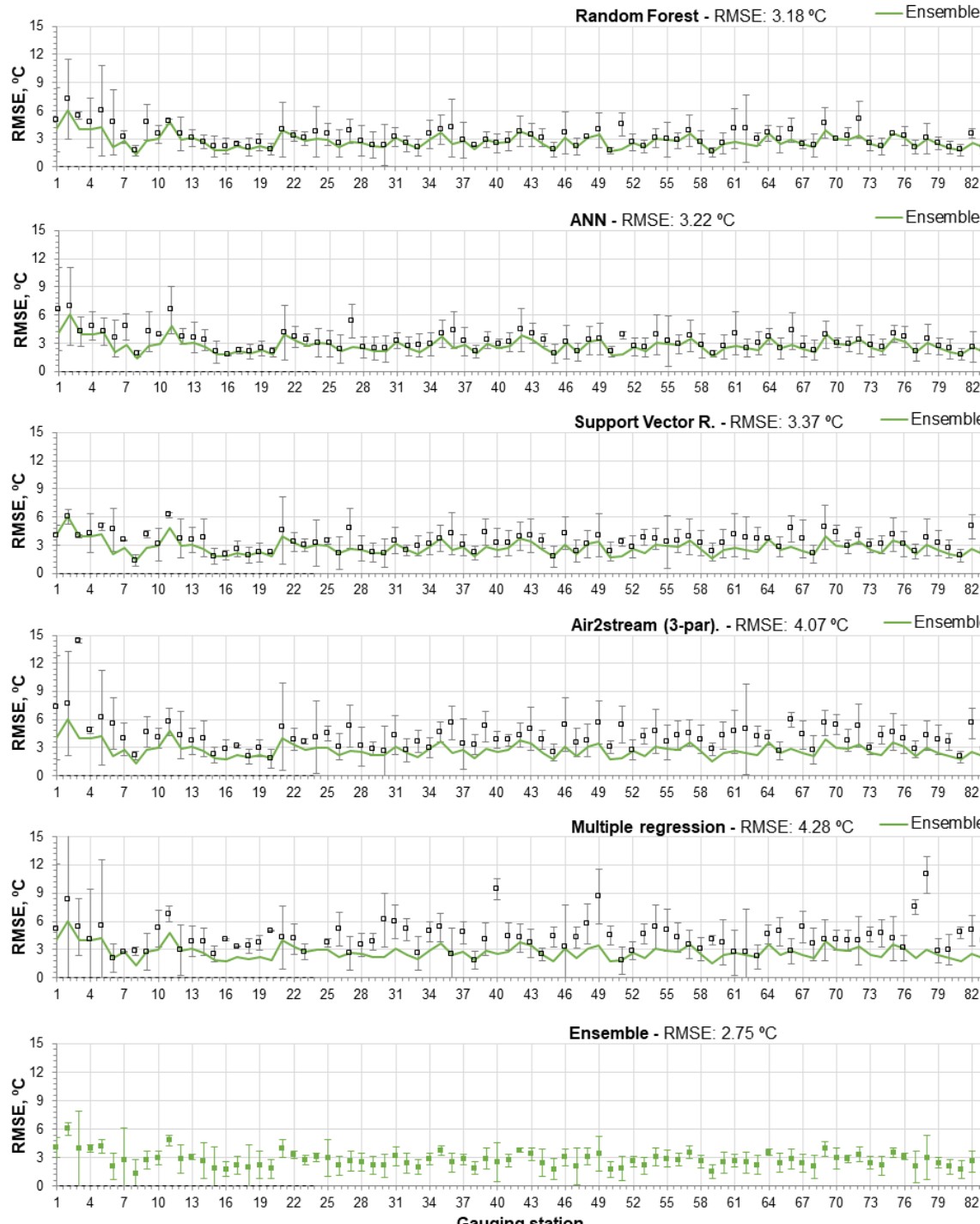

**Figure 5: Root-mean-square error between observed and predicted WT values obtained during the testing phase with all models (with standard deviation of interannual RMSE), considering the model results and the ensemble of all models results. Stations are ordered by the number of testing dataset values, from smallest to largest**

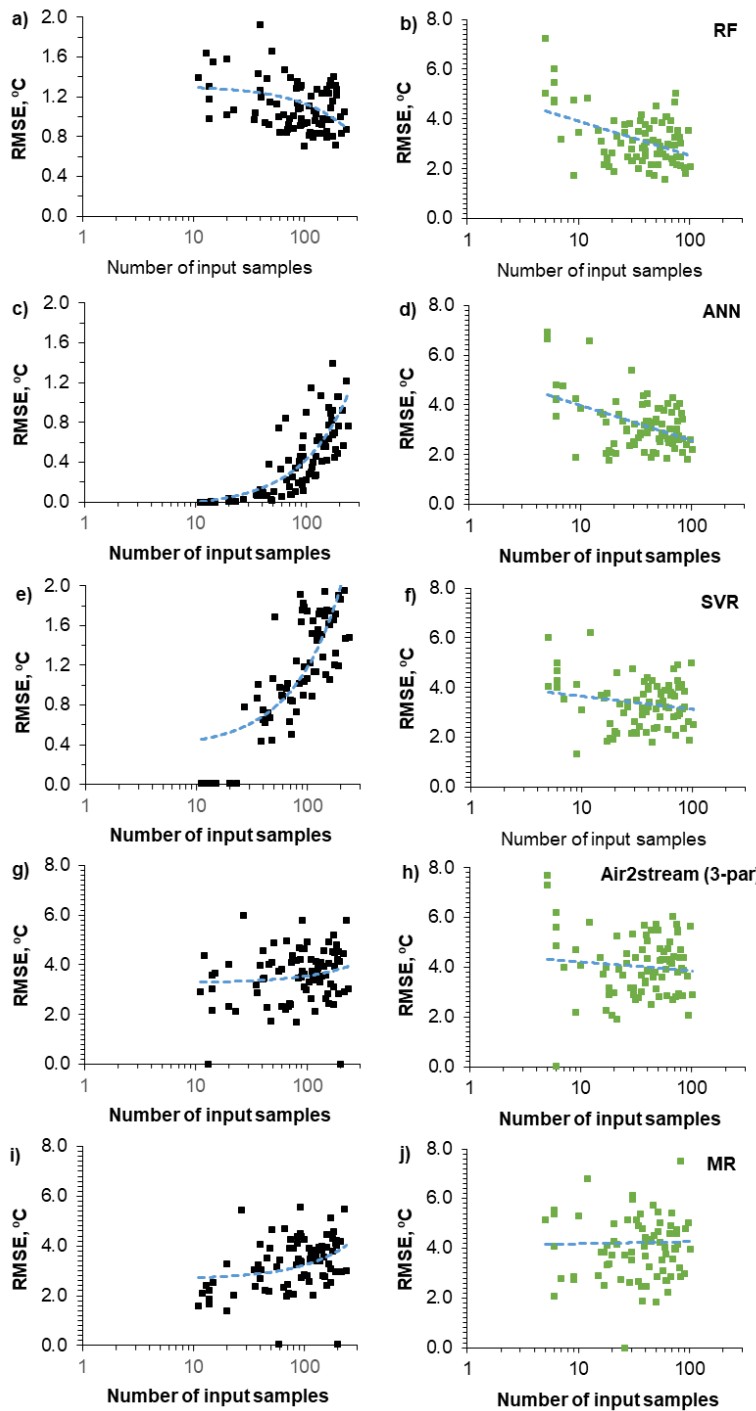

**Figure 6: Root-mean-square error between observed and predicted WT values obtained with all models during the training (black dots) and testing (green dots) phases, ordered by the number of values in the training and testing datasets (from smallest to largest)**

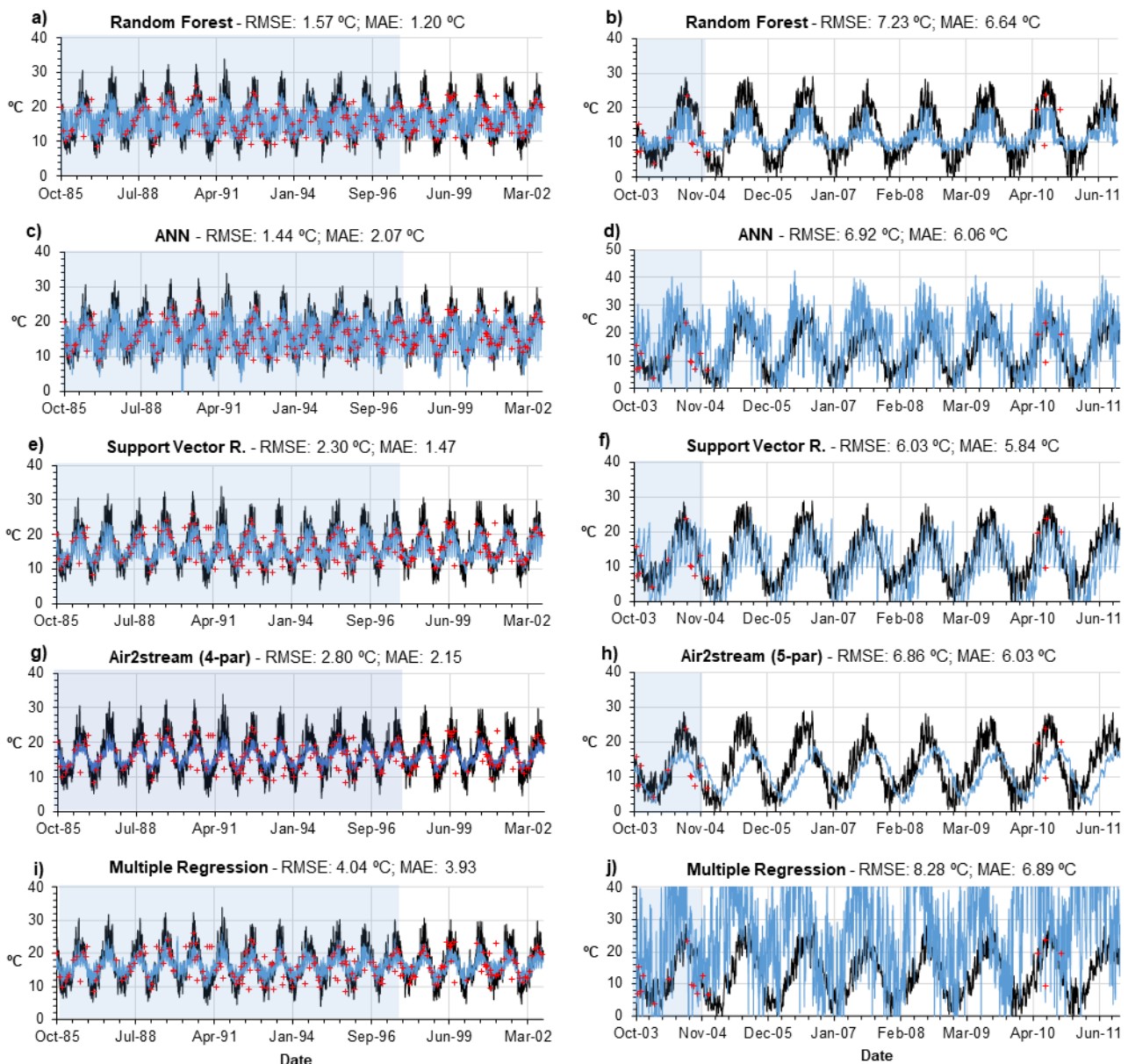

**Figure 7: Root-mean-square error between observed (Red dots) and predicted WT (Blue line) values obtained during the calibration (blue shadow area) and testing phase (white shadow area) with all models for Station 59 (graphs on left) and Station 2 (graphs on right). Air temperature (black line)**

**4.3 Model intercomparison – seasonal datasets**

The results obtained for the dry and wet season testing datasets, considering all metrics, suggest that models performance is better for the dry season, with the exception of the results obtained with the Air2stream model using 3, 4 and 5 parameters (Tables A4 and A5). The model using 3 and 4 parameters does not consider the effect of river discharge and the 5-parameter version assumes that the effect of the discharge can be retained using only a constant value. This suggests that the inclusion of discharge data increased the error in the wet season simulation for the Air2stream model with 7 and 8 parameters. Following

the initial selection of the gauging and water quality stations, the missing discharge values were replaced by the corresponding climatological year value. Missing discharge data replacement varied from 0.0% to 82.6% ($\mu$=30.0; $\sigma$=22.3). Approximately 28% of the stations have missing discharge values of over 50%, which represents an important source of uncertainty that probably affected the Air2stream model performance.

The results obtained with the best performing model (RF) considering the annual datasets are in line with the previous

conclusion that model performance is better for the dry season, but only when the DOY predictor is excluded. The inclusion of the DOY predictor modified the correlation among the different variables and the performance of the models over the wet- and dry season, enhancing the importance of this variable in relation to the overall modeling performance.

Overall, the results are, as expected, similar to those obtained for the annual datasets, showing that the ANN and the SVR models overfitted the training datasets, in particular during the wet season, which also contributed to the worst model

performance during this season. The differences regarding the mean MAE and RMSE of the testing phase are very small among the ML models, with the results of the ANN ensemble coming out slightly ahead of those obtained through the RF and SVR ensemble for both seasons. This deviation in terms of the results obtained for the annual datasets is driven by the difference in the length of the annual versus seasonal datasets. and, consequently, the computation of the metrics, namely the MAE and the RMSE, highlighting the similarity between the ML models results. This is further emphasized by the mean NSE

and KGE values, which, in the case of the wet season testing datasets, provide a contradictory result. According to the mean NSE, the RF and SVR model ensemble produce the best results (NSE: RF: 0.13 (±1.91); SVR: 0.13 (±1.10); ANN: 0.10 (±1.22)), nonetheless the mean KGE values favor the ANN ensemble over the other ML results (KGE: RF: 0.46 (±0.26); SVR: 0.37 (±0.26); ANN: 0.48 (±0.36)). The Air2stream model with 3-parameters is the best of the hybrid model parameterizations followed by the MR (Tables A4 and A5).

## 4.4 Modified training datasets – Over/undersampling technique

Based on the result obtained in section 4.2, the best performing model was the RF and this model was, therefore, considered to evaluate the improvement of accuracy driven by the modified training datasets. The training datasets derived from the application of SMOGN and the ML optimization modeling approach have different characteristics due to the differences in the degree of over/undersampling conducted ("Extreme" versus "Balanced") and the selection of the domain region of WT values considered to be rare ("High", "Both", "Low"). Table 6 shows the number of values in the raw training datasets and in the training datasets obtained with SMOGN corresponding to the best RF model performance. In general terms, considering all stations, oversampling had a more pronounced effect on 50% of the stations (6 stations), while undersampling influenced the sampling process in the case of 33% of the stations. For the remaining two stations over and undersampling had an identical effect on the total number of raw training datasets. The Extreme/Both parameterization was considered for the modeling of 58% of the stations (7 stations), suggesting that more over/undersampling was the best modeling solution for both, high and low regions. This parameterization was followed by the Extreme/High and Balance/Both, with 25% (3 stations) and 17% (2 stations), respectively (Table A8). The WT range affected by both the over and undersampling process was similar for stations 46, 60, 73, as described by the mean WT values (Table 6). These results suggest the tendency for stations with a lower number of values to be affected in different WT ranges, a fact mainly driven by the availability of samples within each region of the response variable WT.

The results obtained with the RF model forced with the modified training datasets had a significant effect on the modeling results. The RF model performance considering the raw training datasets and the modified training datasets is shown in Tables A6 and A7, respectively. The mean RMSE and MAE values obtained between the predicted and observed datasets were reduced from 0.0 to 21.9% ($\mu=8.6\pm8.2$) and from 0.0 to 29.9% ($\mu=10.3\pm9.2$), which can be considered a significant improvement of the RF model accuracy (Table 7). In fact, the RMSE and MAE values were reduced by more than 18% and 15%, respectively for 50% of the stations with less than 80 training samples.

**Table 6: Number of values in the raw training datasets and the training datasets obtained with SMOGN, corresponding to the best RF model performance**

| Station | Train (Raw datasets) (number of values) | Oversampling | | Undersampling | | Train (Modified datasets) (number of values) | Test (number of values) |
|---|---|---|---|---|---|---|---|
| | | Number of values | Water temperature range, ºC (minimum; maximum; average with standard deviation) | Number of values | Water temperature range, ºC (minimum; maximum; average with standard deviation) | | |
| 1 | 10 | 0 | - | 0 | - | 10 | 5 |
| 7 | 14 | 15 | 16.00; 28.00; 19.27±3.39 | 4 | 19.00; 28.00; 22.25±4.27 | 25 | 7 |
| 12 | 35 | 4 | 12.94; 15.75; 14.20±1.40 | 10 | 17.60; 22.40; 20.10±1.54 | 29 | 15 |
| 13 | 35 | 16 | 16.13; 19.80; 17.54 ±1.15 | 10 | 8.30; 15.30; 13.16 ±2.28 | 41 | 16 |
| 22 | 50 | 39 | 8.00; 26.00; 16.36±4.51 | 17 | 15.00; 26.00; 19.76±2.95 | 72 | 22 |
| 29 | 69 | 23 | 16.75; 22.56; 18.93±1.57 | 23 | 6.50; 16.00; 12.42±2.54 | 69 | 30 |
| 30 | 71 | 41 | 8.87; 22.00; 16.60±4.47 | 9 | 7.00; 22.00; 14.79±4.84 | 103 | 31 |
| 46 | 98 | 12 | 8.20; 36.00; 19.21±8.58 | 33 | 14.60; 22.00; 18.58 ±2.06 | 77 | 43 |
| 59 | 137 | 120 | 8.60; 26.00; 17.91±3.53 | 40 | 9.00; 23.00; 15.77±4.23 | 217 | 60 |
| 60 | 141 | 27 | 10.03; 25.03; 16.64±5.76 | 31 | 14.10; 21.20; 17.16±2.18 | 137 | 61 |
| 73 | 177 | 15 | 8.52; 30.00; 16.79±6.41 | 53 | 13.50; 19.10; 17.04±1.22 | 139 | 76 |
| 83 | 236 | 353 | 8.50; 28.00; 16.96±3.41 | 32 | 17.00; 26.00; 20.18±2.19 | 557 | 102 |

**Table 7: Percentual variation between the metrics obtained with the raw training datasets and the modified training datasets with RF model**

| Annual | TRAIN | | | | | | TEST | | | | | |
|---|---|---|---|---|---|---|---|---|---|---|---|---|
| Station/metric | MAE (%) | RMSE (%) | NSE (%) | KGE (%) | bias (%) | R$^2$ (%) | MAE (%) | RMSE (%) | NSE (%) | KGE (%) | Bias (%) | R$^2$ (%) |
| 1 | -153.6 | -129.0 | 19.7 | 4.5 | 100.0 | 22.1 | 2.5 | 0.7 | -217.0 | -10.1 | 19.5 | 1.2 |
| 7 | 7.3 | -51.6 | 41.3 | 16.0 | 100.0 | 45.7 | 18.0 | 18.0 | -38.8 | -28.5 | -3185.7 | -40.2 |
| 12 | 100.0 | 100.0 | -8.9 | -23.9 | 100.0 | -5.8 | 14.8 | 7.8 | -82.2 | 34.4 | 65.4 | -17.1 |
| 13 | -17.5 | -25.8 | 8.5 | -6.8 | 197.5 | 13.4 | 29.2 | 17.6 | -77.2 | -150.9 | 406.2 | -26.5 |
| 22 | 92.6 | 89.9 | -10.1 | -24.5 | 100.0 | -6.4 | 3.6 | 2.2 | -5.2 | 4.9 | 0.3 | -8.2 |
| 29 | 22.7 | 18.3 | -1.4 | -10.5 | 100.0 | -0.4 | 7.5 | 21.9 | -23.7 | -40.6 | -55.2 | -20.0 |
| 30 | 60.6 | 37.8 | -6.6 | -15.9 | 100.0 | -5.2 | 5.9 | 3.2 | -8.1 | -11.9 | 60.9 | -1.1 |
| 46 | -60.7 | -79.6 | 3.6 | 1.8 | 1168.2 | 3.2 | 23.1 | 18.7 | -23.1 | -20.4 | 390.2 | -26.5 |
| 59 | 20.9 | 5.5 | 0.0 | -0.3 | 187.9 | 0.1 | 9.2 | 8.7 | -3.1 | -1.0 | 67.6 | -2.3 |
| 60 | -3.7 | -10.2 | -0.2 | -0.5 | 53.4 | -0.1 | 8.4 | 3.0 | -2.5 | -7.3 | 29.5 | -0.3 |
| 73 | 0.0 | 0.0 | 0.0 | 0.0 | 0.0 | 0.0 | 1.0 | 0.0 | 0.0 | 0.0 | 0.0 | 0.0 |
| 83 | 0.1 | -20.7 | 3.8 | 3.5 | 263.2 | 3.5 | 0.4 | 1.2 | -1.1 | -2.1 | -161.8 | -2.0 |
| Average | **5.7 (±67.7)** | **-5.5 (±64.9)** | **4 .2 (±14.2)** | **-4.7 (±12.1)** | **205.9 (±310.8)** | **5.8 (±15.0)** | **10.3 (±9.2)** | **8.6 (±8.2)** | **-40.2 (±62.6)** | **-19.5 (±45.4)** | **-196.9 (±955.5)** | **-11.9 (±13.8)** |
| **Maximum** | 100 | 100 | 41 | 16 | 1168 | 46 | 29.2 | 21.9 | 0 | 34 | 406 | 1 |
| **Minimum** | -154 | -129 | -10 | -25 | 0 | -6 | 0.0 | 0.0 | -217 | -151 | -3186 | -40 |

**4.5 Feature importance**

Table 8 shows the mean feature importance obtained with the best performing model (Random Forest Regressor, Pedregosa et al., 2011) considering the mean annual RMSE, a RF with the following parameters: n_estimators = 50; max_depth = 485; min_samples_split = 5; max_features = 'auto'; bootstrap = True; random_state = 42; considering all stations datasets. The maximum importance values show that all features are relevant, at least for some stations, and that they should not be discarded. The mean importance values indicate that the mean air temperature and the DOY are of upmost importance in relation to the model training process, followed by the maximum and minimum air temperature. Discharge, global radiation and MOY clearly play a secondary role, as described by the mean and standard deviation values. Table A9 shows the evaluation of the RF model performance during the training and testing phases considering the annual datasets and the sequential increase of the model predictors. The results show that, on average, the inclusion of all predictor variables have a significant effect on model performance.

**Table 8: Mean input feature importance obtained with a Random Forest regressor**

|  | Mean Air temperature | Maximum air temperature | Minimum air temperature | Discharge | Global radiation | Month of the year | Day of the year |
|---|---|---|---|---|---|---|---|
| **Mean** | 0.20 | 0.12 | 0.15 | 0.09 | 0.10 | 0.06 | 0.29 |
| **Standard deviation** | 0.16 | 0.10 | 0.10 | 0.09 | 0.07 | 0.07 | 0.20 |
| **Maximum** | 0.70 | 0.46 | 0.62 | 0.33 | 0.28 | 0.34 | 0.82 |
| **Minimum** | 0.02 | 0.01 | 0.02 | 0.01 | 0.01 | 0.00 | 0.01 |

## 4.6 Effect of the watershed time of concentration on model performance

The results suggest that, tendentially, there are more training and testing datasets available for the largest watersheds (Fig. 8a and 8b) and that the watershed time of concentration increases with the watershed area according to a power law (Fig. 8c). Additionally, the graphic correlation of the RMSE between the observed river WT and the predicted WT (Training datasets) obtained with the best performing model run - the RF ensemble model and the best individual RF run with the watershed time of concentration - revealed the existence of a very specific linear pattern within the dataset (Fig. 9a and 9b). After the datasets z-score normalization and the application of the Gaussian Mixture Models algorithm with the following parameters: n_components=2, covariance_type='diag',init_params='random',warm_start=True (*vide* Pedregosa et al., 2011), two different data samples were extracted. This small set of values, 19 (watershed area: $\mu$= 106 km$^2$; $\sigma$=153) (Fig. 9a) and 19 (watershed area: $\mu$= 106 km$^2$; $\sigma$=153) (Fig. 9b) corresponds to 35% of the stations with fewer than 125 training values, a fact that enhances the non-random nature of this correlation. This correlation shows how the RMSE obtained with the RF increases with the watershed area, clearly showing the significant effect upstream conditions have on river WT. The RMSE increases by an average of 0.1 ºC with a one-hour increase in the watershed time of concentration, considering the RF ensemble aggregation approach (Fig. 9b).

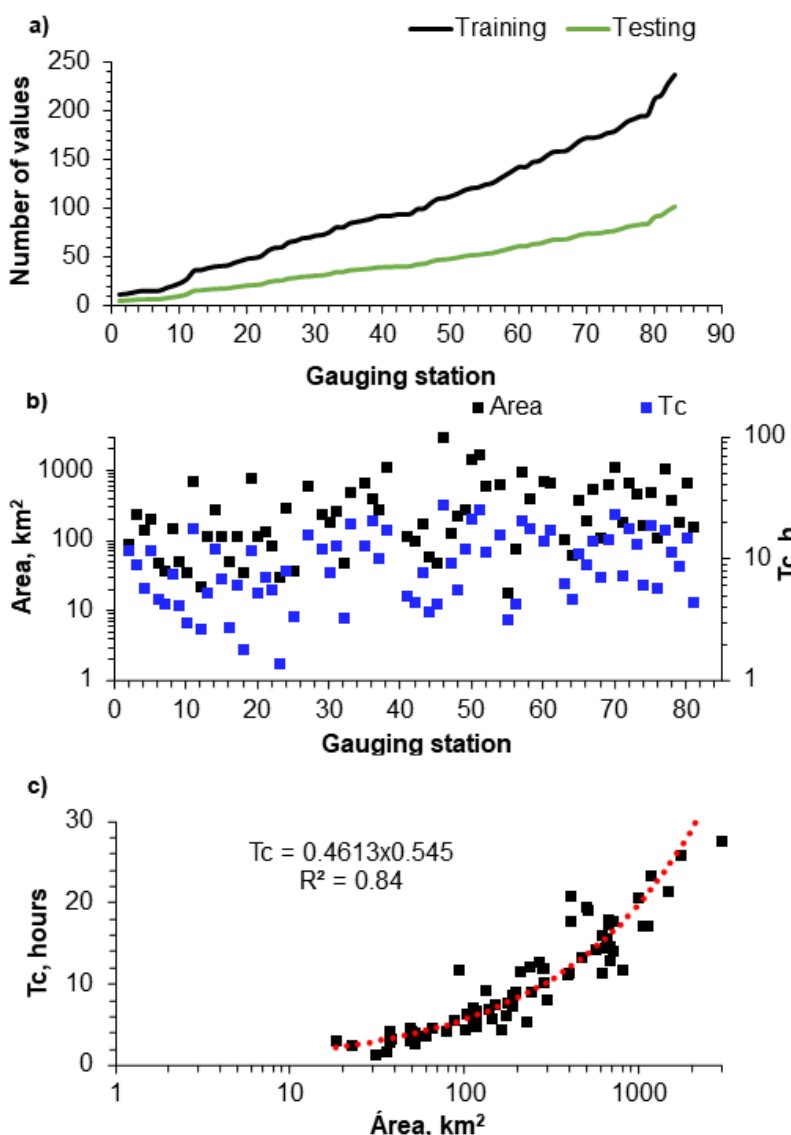

**Figure 8: a) Number of training and testing datasets of each station. b) Watershed time of concentration and area of each station. c) Watershed time of concentration versus watershed area**

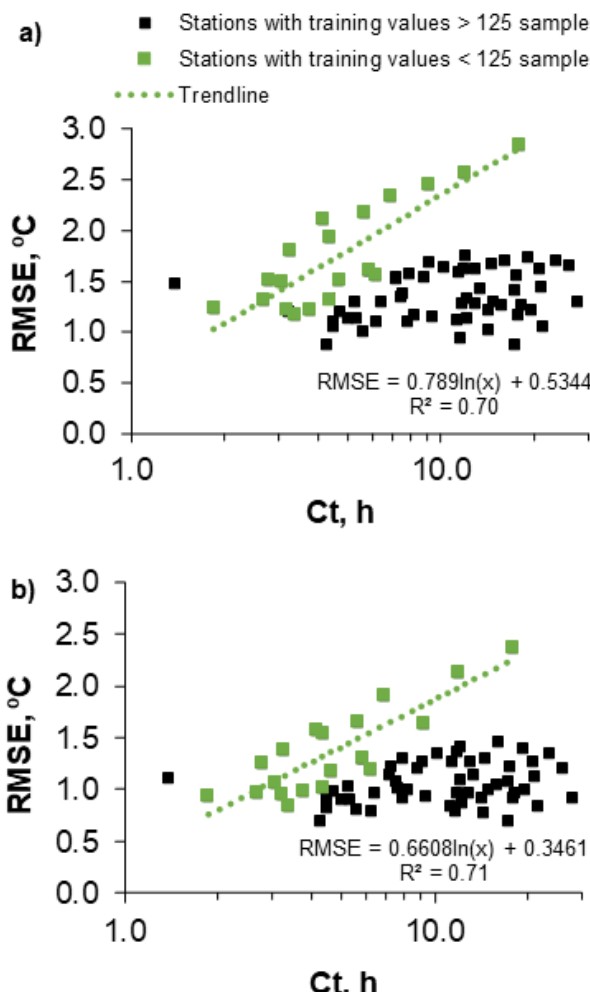

Figure 9: a) RMSE between observed and simulated river WT with the Random Forest best model run versus the watershed time of concentration. b) RMSE between observed and simulated river WT with the Random Forest ensemble aggregation approach versus the watershed time of concentration (Ct).

**5 Discussion**

Overall, the results of the model's ensemble (mean RMSE: 2.75 ºC; σ = 1.00) driven mainly by the ML algorithms predictions are in line with the results obtained in other studies, namely Rabi et al., (2015) (ANN - RMSE: μ=2.06 ºC) and Zhu et al., (2019a) (MR – RMSE: 2.74 ºC). This is quite significant considering the scale of the missing training and testing datasets corresponding to this study (μ= 98.8%; σ=0.68). These results are, as expected, worse than the results obtained in some of the more recent studies in which ML algorithms were used to predict river WT (Table1). However, the availability of training data for most of these studies was impressively good in terms of quantity and quality, which is, of course, reflected in the overall results.

The selection of the best approach to model river WT is not an easy task, as ML algorithm performance levels are very similar (Feigl et al., 2021). That said, considering all the metrics, the RF model ensemble produced the best results for the annual datasets and was the model that provided the greatest contribution in relation to overall ensemble results. As such, this was selected as the best model for modeling river WT for stations with limiting forcing data. However, this is not in line with the findings of other studies. Rajesh and Rehana (2021) and Rehana (2019) concluded that the SVR model was the most robust model for predicting river WT temperature on a daily timescale. Feigl et al., (2021) concluded that the FNNs and the recurrent neural networks (RNNs) performed better than the RF model. It is, however, important to highlight the significant variations in terms of the number of watersheds studied and the overall length of the training datasets used across all the different studies, which effectively could explain the different findings in relation to model performance.

One of this study's most significant conclusions is that, from a practical point of view, the application of all the models considered in this study is relevant. In fact, our results show that all models considered were best performers for some of the station datasets, including the MR, which was the best model for 14 stations. The results show that the advantages of the state-of-the-art ML models and the Air2stream model are reduced when the training datasets are very small (<200 values) and spans a long period of time. The information contained in the training datasets is not sufficient for the definition of the unknown underlying function that best relates the input variables to the output variable. Hence the less complex approaches, such as MR, may surpass the results produced by ML algorithms.

The ML algorithms can considerably improve on the prediction results produced by the current state-of-the-art Air2stream model, regardless of the model parametrization. This finding concurs with that of Feigl et al. (2021) but is contrary to the results of the study carried out by Zhu et al. (2019d), which assessed the performance of a suite of machine-learning models for daily stream WT. However, in the case of our study the performance of the Air2stream model was affected by the missing training data, namely, the discharge datasets, which proved to be a significant obstacle for this model. When the dataset gap is very large, the structure of the Air2stream model with 6 or more parameters may become very complex when compared to the

number of observed WT values, increasing the risk of overfitting (Piccolroaz, 2016). This explains the fact that, considering all the metrics, the best results were obtained with the 3-parameter model, the simplest version of the Air2stream model. The

3-parameter model does not consider the river discharge and depth on a daily timescale and, as such, can be successfully applied if the longitudinal gradient of temperature is small (Toffolon and Piccolroaz, 2015). The results of our study correspond to those obtained by Piccolroaz (2016) regarding the effect of missing data during the modeling of the WT of two lakes located in the USA (Lake Erie and Lake Superior) with the 4- and 6-parameter Air2stream model. When the length of the calibration period is of one year and the percentage of missing data is in the range of 99%, the RMSE between observed and predicted

lake WT is >3.5 ℃. It is also relevant to mention that the results of this study suggest that, besides the WT dataset gaps, the modeling results were also affected by the presence of a large number of WT outliers, by the uncertainty induced by the mean air temperature ERA5-Land reanalysis datasets and by upstream conditions, which increases with the watershed area. In terms of input dataset quality, the results of this study suggests that when the missing datasets reach 98%, a RMSE <3.0℃ is indicative of a good modeling performance. Importantly, this error can also be further decreased by the generation of synthetic

samples to some poorly represented ranges within the datasets by applying a model such as SMOGN (Branco et al., 2017).

The success of the models considered in this study, namely the ML algorithms, is undoubtedly linked to the hyperparameter optimization algorithm, a conclusion that is in line with the findings of Feigl et al. (2021). The feature importance analysis showed that all the predictors (mean, max. and min. daily air temperature, mean daily total radiation, discharge, MOY and DOY) are relevant to model performance, a conclusion that also concurs with the findings of Feigl et al. (2021). Nonetheless

the results highlight the importance of the daily mean air temperature and DOY. The DOY was the most relevant variable. In fact, the inclusion of the DOY modified the correlation among the different variables and the performance of the models across the wet and dry season, increasing the importance of this variable to the overall modeling performance, which is in line with the findings of Zhu et al. (2019d). This suggests that the correlation associated with the other input variables and the observed river WT is, in fact, rather weak, which relates to the length and quality of the training datasets, as well as the uncertainty

caused by the fact that a river's upstream environmental conditions can have a significant effect on WT predictions. However, it is also worth mentioning the lack of clarity in relation to the exact extent of the upstream area controlling the river energy balance at a given point (Moore et al., 2005) and, as such, the averaging of the predictor variables over the watershed area might not be the best solution. There are a number of limitations associated with our study that should be addressed in future studies. Firstly, the fact that, regardless of the hyperparameter optimization and the inclusion of regularization and dropout

layers to minimize overfitting in the ANN model, the results show that when the training datasets contain fewer than 30 values, the model will considerably overfit the datasets and considerably reduce the model's predictive capacity. This limitation might be minimized with more effective control of the number of training epochs and the regularization algorithm. It is also important to mention the fact that the hyperparameter optimization algorithm was not applied to all the station datasets, hence the ML

algorithms might be further improved. Due to the lack of physical restraints, ML models might fail when extrapolating outside the range of their training datasets. This was not fully evaluated in this study due to the number of watersheds studied but certainly requires further investigation in the future. The modeling of 100 synthetic training datasets per station with the RF model to evaluate the SMOGN algorithm performance was very time-consuming. In fact, the average time required to model each station considered was 4.0 hours $\pm$ 0.45. Therefore, the accuracy of the RF model can probably be further increased if the number of training datasets is higher. If possible, this sensitive analysis should be combined with the evaluation of the loss of quality/consistency of the training datasets due to undersampling. The results of this study demonstrate the feasibility of finding a correlation between the prediction error between observed and predicted river WT values and the watershed time of concentration. However, the number of samples that form this correlation is small (19) and, as such, the number of watersheds studied needs to be increased to strengthen this correlation and scale it to other watersheds. The inclusion of the watershed soil type as a predictor variable would also be of relevance. It is also important to note that the results of this study are restricted to the Mediterranean region and, therefore, the expansion of the study area to other latitudes to consider different climate and soil conditions would also be interesting, namely, the north of Europe and Africa where data scarcity is quite relevant.

## 5 Conclusion

The results obtained with this study demonstrate, from a practical modeling perspective, the validity of applying all the models considered in this study - Random Forest, Artificial Neural Network, Support Vector Regression, Air2stream, and Multiple Regression - when the number of predictor variables and observed river WT values is limited. It is also of upmost importance to optimize the ML algorithms hyperparameters. The Tree-structured Parzen Estimators algorithm has proved to be a good solution. The results of this study also show the viability of using all available predictor variables and highlights the importance of the day of the year and the mean daily air temperature. Regardless of the greater degree of modeling performance that can be attained with an ensemble of all the different models, the Random Forest model with the following parameters: n_estimators = 50; max_depth = 485; min_samples_split = 5; max_features = 'auto'; bootstrap = True; random_state = 42, produces the best performance and may represent an effective solution for modeling river WT with limiting forcing data. Importantly, the accuracy of the Random Forest model can be significantly improved by the generation of synthetic samples to some poorly represented ranges within the datasets by applying a model such as SMOGN (Branco et al., 2017).

It is also relevant to mention that a logarithmic correlation exists in relation to the RMSE between the observed and predicted river WT and the watershed time of concentration. The RMSE increases by an average of 0.1 ºC with a one-hour increase in the watershed time of concentration (watershed area: $\mu$= 106 km$^2$; $\sigma$=153), a conclusion that may prove useful for increasing

our understanding of the effects of catchment size and landscape on runoff generation and, consequently, on river energy balance.

**Appendix A**

**Table A1: Model parameters and optimization range**

| Model | Prior distribution | Parameter | Optimization range |
|-------|-------------------|-----------|-------------------|
| | uniform | 'n_estimators' | [50, 2000] |
| | uniform | 'max_depth' | [10, 1000] |
| RF | uniform | 'min_samples_split' | [2, 10] |
| | - | 'max_features' | [auto, sqrt] |
| | - | 'bootstrap' | [True, False] |
| | categorical | 'n_layers' | [1, 2] |
| | uniform integer | 'n_units_layer' | [10, 50] |
| | categorical | 'act_func_type' | ['Relu', 'PRelu', 'Elu', 'Tanh', 'Sigmoid'] |
| | categorical | 'regularization' | [True, False] |
| ANN | quantized distribution | 'n_epochs' | With regularization: [500, 1000]; without regularization: [20, 300] |
| | uniform | 'dropout' | [0, 1.0] |
| | loguniform | 'batch_size' | [5, 20] |
| | uniform | 'initial_value' | [0.001, 0.1] |
| | uniform | 'reduction_freq' | [10, 200] |
| | uniform | 'decay_rate' (regularization) | [0.0001, 0.001] |
| | Categorical | 'C' | [0.1,1,100,1000] |
| | Categorical | 'kernel' | ['rbf','poly','sigmoid','linear'] |
| SVR | Categorical | 'degree' | [1,2,3,4,5,6] |
| | Categorical | 'gamma' | [1, 0.1, 0.01, 0.001, 0.0001] |
| | Categorical | 'epsilon' | [0.0001, 0.0005, 0.001, 0.005, 0.01, 0.05, 0.1, 0.5, 1, 5, 10] |

**Table A2: Description and optimization range (SMOGN, 2022) of modeling parameters considered for the application of SMOGN.**

| Parameter | Description | Parameter search space |
|---|---|---|
| k | Specifies the number of neighbors to consider for interpolation used in oversampling | uniform [1, 10] |
| samp_method | If 'balance' is specified, less over/under-sampling is applied. If 'extreme' is specified, more over/under-sampling is applied | Categorical [extreme, balance] |
| rel_thres | Specifies the threshold of rarity, takes a real number between 0 and 1. | uniform [0, 1] |
| rel_coef | Corresponds to the box plot coefficient used to automatically determine extreme and therefore rare "minority" values in y, when rel_method = 'auto' | uniform [0.01, 0.4] |
| rel_method | rel_method argument takes a string, either 'auto' or 'manual'. It specifies how relevant or rare "minority" values in y are determined. If 'auto' is specified, "minority" values are automatically determined by box plot extremes. | 'auto' |
| rel_xtrm_type | The rel_xtrm_type argument takes a string, either 'low' or 'both' or 'high'. It indicates which region of the response variable y should be considered rare or a "minority", when rel_method = 'auto' | Categorical [high, both, low] |

**Table A3:** Evaluation of model performance during the training and testing phases considering the annual datasets. Mean MAE, RMSE, NSE, KGE, bias and $R^2$ (with standard deviation) between observed and predicted WT values

| Annual | TRAIN | | | | | |
|---|---|---|---|---|---|---|
| Model/metric | MAE (ºC) | RMSE (ºC) | NSE | KGE | bias (ºC) | $R^2$ |
| RF | 0.86 (±0.25) | 1.13 (±0.30) | 0.93 (±0.03) | 0.85 (±0.07) | -0.01 (±0.06) | 0.96 (±0.02) |
| ANN | 0.29 (±0.29) | 0.44 (±0.40) | 0.99 (±0.03) | 0.98 (±0.03) | 0.01 (±0.02) | 0.99 (±0.03) |
| SVR | 0.82 (±0.54) | 1.19 (±0.64) | 0.91 (±0.06) | 0.88 (±0.09) | 0.00 (±0.11) | 0.92 (±0.05) |
| Air2stream (3-par) | 2.82 (±0.86) | 3.65 (±0.96) | 0.33 (±0.25) | 0.33 (±0.32) | 0.01 (±0.01) | 0.33 (±0.25) |
| Air2stream (4-par) | 2.83 (±0.86) | 3.65 (±0.97) | 0.33 (±0.25) | 0.34 (±0.31) | 0.00 (±0.01) | 0.33 (±0.25) |
| Air2stream (5-par) | 2.72 (±0.88) | 3.54 (±0.98) | 0.36 (±0.25) | 0.38 (±0.29) | 0.00 (±0.01) | 0.36 (±0.25) |
| Air2stream (7-par) | 2.67 (±0.86) | 3.50 (±0.99) | 0.38 (±0.25) | 0.42 (±0.28) | 0.01 (±0.02) | 0.38 (±0.25) |
| Air2stream (8-par) | 2.68 (±0.87) | 3.49 (±0.99) | 0.39 (±0.24) | 0.43 (±0.28) | 0.01 (±0.04) | 0.39 (±0.24) |
| MR | 2.55 (±0.79) | 3.33 (±0.95) | 0.47 (±0.27) | 0.49 (±0.24) | 0.00 (±0.00) | 0.44 (±0.22) |
| **Ensemble** | **0.28 (±1.07)** | **0.41 (±1.36)** | **0.99 (±0.32)** | **0.98 (±0.27)** | **0.01 (±0.04)** | **0.99 (±0.30)** |
| | | | | | | |
| Annual | TEST | | | | | |
| Model/metric | MAE (ºC) | RMSE (ºC) | NSE | KGE | bias (ºC) | $R^2$ |
| RF | 2.44 (±0.91) | 3.18 (±1.06) | 0.52 (±0.23) | 0.60 (±0.20) | -0.07 (±1.11) | 0.60 (±0.18) |
| ANN | 2.50 (±0.86) | 3.22 (±1.05) | 0.48 (±0.28) | 0.66 (±0.18) | -0.12 (±0.94) | 0.55 (±0.22) |
| SVR | 2.60 (±0.86) | 3.37 (±0.96) | 0.47 (±0.19) | 0.53 (±0.21) | 0.00 (±0.83) | 0.54 (±0.18) |
| Air2stream (3-par) | 3.17 (±1.06) | 4.07 (±1.18) | 0.21 (±0.32) | 0.29 (±0.32) | -0.18 (±1.15) | 0.34 (±0.22) |
| Air2stream (4-par) | 3.30 (±1.15) | 4.24 (±1.37) | 0.11 (±0.73) | 0.30 (±0.29) | -0.04 (±1.30) | 0.32 (±0.23) |
| Air2stream (5-par) | 3.53 (±1.08) | 4.37 (±1.13) | 0.06 (±0.59) | 0.18 (±0.38) | -0.12 (±1.03) | 0.30 (±0.22) |
| Air2stream (7-par) | 3.74 (±1.15) | 4.73 (±1.36) | -0.13 (±0.81) | 0.19 (±0.32) | -0.50 (±1.51) | 0.24 (±0.22) |
| Air2stream (8-par) | 3.94 (±1.35) | 5.06 (±1.73) | -0.56 (±2.27) | 0.16 (±0.44) | -0.42 (±1.65) | 0.23 (±0.22) |
| MR | 3.34 (±1.29) | 4.28 (±1.62) | 0.32 (±0.34) | 0.36 (±0.27) | -0.46 (±2.14) | 0.34 (±0.22) |
| **Ensemble** | **2.14 (±0.83)** | **2.75 (±1.00)** | **0.56 (±0.48)** | **0.61 (±0.25)** | **-0.16 (±0.73)** | **0.60 (±0.18)** |

**Table A4: Evaluation of model performance during the training and testing phases considering the dry season datasets. Mean MAE, RMSE, NSE, KGE, bias and $R^2$ (with standard deviation) between observed and predicted WT values**

| Dry season | TRAIN | | | | | |
|---|---|---|---|---|---|---|
| Model/metric | MAE (ºC) | RMSE (ºC) | NSE | KGE | bias (ºC) | $R^2$ |
| RF | 0.87 (±0.28) | 1.13 (±0.34) | 0.91 (±0.09) | 0.83 (±0.88) | 0.09 (±0.28) | 0.95 (±0.02) |
| ANN | 0.33 (±0.30) | 0.47 (±0.41) | 0.98 (±0.03) | 0.97 (±0.03) | 0.01 (±0.03) | 0.98 (±0.03) |
| SVR | 0.84 (±0.54) | 1.20 (±0.68) | 0.89 (±0.07) | 0.86 (±0.10) | 0.07 (±0.15) | 0.91 (±0.06) |
| Air2stream (3-par) | 2.93 (±0.95) | 3.67 (±1.08) | 0.21 (±0.25) | 0.23 (±0.34) | 0.30 (±0.45) | 0.26 (±0.25) |
| Air2stream (4-par) | 2.96 (±0.93) | 3.69 (±1.06) | 0.21 (±0.25) | 0.23 (±0.32) | 0.37 (±0.52) | 0.27 (±0.25) |
| Air2stream (5-par) | 2.81 (±0.95) | 3.55 (±1.07) | 0.25 (±0.24) | 0.23 (±0.31) | 0.04 (±0.19) | 0.28 (±0.24) |
| Air2stream (7-par) | 2.80 (±0.92) | 3.55 (±1.05) | 0.27 (±0.24) | 0.29 (±0.30) | 0.13 (±0.28) | 0.29 (±0.23) |
| Air2stream (8-par) | 2.82 (±0.92) | 3.55 (±1.04) | 0.27 (±0.24) | 0.30 (±0.30) | 0.19 (±0.32) | 0.29 (±0.24) |
| MR | 2.55 (±0.80) | 3.22 (±0.96) | 0.37 (±0.27) | 0.39 (±0.24) | 0.13 (±0.19) | 0.41 (±0.22) |
| **Ensemble** | **0.31 (±1.12)** | **0.44 (±1.37)** | **0.98 (±0.37)** | **0.97 (±0.33)** | **0.01 (±0.22)** | **0.98 (±0.34)** |
| Dry season | TEST | | | | | |
| Model/metric | MAE (ºC) | RMSE (ºC) | NSE | KGE | bias (ºC) | $R^2$ |
| RF | 2.37 (±1.17) | 3.01 (±1.30) | 0.33 (±0.62) | 0.55 (±0.24) | 0.29 (±1.55) | 0.57 (±0.22) |
| ANN | 2.19 (±0.93) | 2.80 (±1.10) | 0.31 (±0.71) | 0.57 (±0.31) | 0.01 (±1.03) | 0.54 (±0.22) |
| SVR | 2.39 (±0.95) | 3.02 (±1.06) | 0.37 (±0.34) | 0.50 (±0.22) | 0.29 (±1.04) | 0.52 (±0.22) |
| Air2stream (3-par) | 3.29 (±1.27) | 4.12 (±1.32) | -0.13 (±0.47) | 0.09 (±0.35) | 0.30 (±1.73) | 0.21 (±0.23) |
| Air2stream (4-par) | 3.65 (±2.57) | 4.49 (±2.51) | -0.28 (±0.87) | 0.11 (±0.34) | 0.79 (±3.20) | 0.24 (±0.26) |
| Air2stream (5-par) | 3.69 (±1.35) | 4.48 (±1.38) | -0.41 (±1.00) | 0.04 (±0.33) | 0.79 (±2.15) | 0.18 (±0.22) |
| Air2stream (7-par) | 3.77 (±2.55) | 4.58 (±2.50) | -0.29 (±0.75) | 0.06 (±0.31) | 0.57 (±3.23) | 0.17 (±0.21) |
| Air2stream (8-par) | 3.97 (±2.66) | 4.84 (±2.67) | -0.59 (±1.78) | 0.06 (±0.35) | 0.75 (±3.36) | 0.18 (±0.22) |
| MR | 3.39 (±2.58) | 4.21 (±2.71) | 0.21 (±0.35) | 0.22 (±0.33) | -0.44 (±3.26) | 0.30 (±0.22) |
| **Ensemble** | **1.98 (±0.96)** | **2.51 (±1.08)** | **0.50 (±0.55)** | **0.63 (±0.28)** | **0.12 (±1.17)** | **0.63 (±0.21)** |

**Table A5: Evaluation of model performance during the training and testing phases considering the wet season datasets. Mean MAE, RMSE, NSE, KGE, bias and $R^2$ (with standard deviation) between observed and predicted WT values**

| Wet season | TRAIN | | | | | |
|---|---|---|---|---|---|---|
| Model/metric | MAE (ºC) | RMSE (ºC) | NSE | KGE | bias (ºC) | $R^2$ |
| RF | 0.84 (±0.27) | 1.11 (±0.33) | 0.91 (±0.06) | 0.80 (±0.09) | -0.07 (±0.15) | 0.94 (±0.04) |
| ANN | 0.25 (±0.28) | 0.37 (±0.40) | 0.98 (±0.04) | 0.98 (±0.04) | 0.01 (±0.03) | 0.98 (±0.03) |
| SVR | 0.75 (±0.53) | 1.06 (±0.66) | 0.91 (±0.07) | 0.88 (±0.10) | -0.03 (±0.17) | 0.92 (±0.06) |
| Air2stream (3-par) | 2.72 (±0.93) | 3.57 (±1.07) | 0.15 (±0.26) | 0.14 (±0.32) | -0.22 (±0.35) | 0.20 (±0.22) |
| Air2stream (4-par) | 2.70 (±0.92) | 3.55 (±1.06) | 0.15 (±0.26) | 0.18 (±0.31) | -0.28 (±0.40) | 0.20 (±0.22) |
| Air2stream (5-par) | 2.64 (±0.95) | 3.48 (±1.11) | 0.20 (±0.25) | 0.18 (±0.29) | -0.02 (±0.16) | 0.23 (±0.24) |
| Air2stream (7-par) | 2.56 (±0.96) | 3.41 (±1.14) | 0.24 (±0.25) | 0.24 (±0.29) | -0.10 (±0.25) | 0.27 (±0.25) |
| Air2stream (8-par) | 2.55 (±0.97) | 3.38 (±1.16) | 0.25 (±0.26) | 0.27 (±0.31) | -0.14 (±0.27) | 0.28 (±0.25) |
| MR | 2.58 (±0.89) | 3.40 (±1.09) | 0.30 (±0.28) | 0.32 (±0.28) | -0.11 (±0.18) | 0.27 (±0.23) |
| **Ensemble** | **0.23 (±1.06)** | **0.35 (±1.37)** | **0.99 (±0.39)** | **0.98 (±0.36)** | **0.01 (±0.19)** | **0.99 (±0.37)** |
| Wet season | TEST | | | | | |
| Model/metric | MAE (ºC) | RMSE (ºC) | NSE | KGE | bias (ºC) | $R^2$ |
| RF | 2.38 (±1.07) | 3.04 (±1.19) | 0.13 (±1.91) | 0.46 (±0.26) | -0.36 (±1.37) | 0.49 (±0.23) |
| ANN | 2.38 (±1.04) | 3.03 (±1.26) | 0.10 (±1.22) | 0.48 (±0.36) | -0.23 (±1.06) | 0.48 (±0.23) |
| SVR | 2.52 (±0.94) | 3.20 (±1.12) | 0.13 (±1.10) | 0.37 (±0.26) | -0.42 (±0.96) | 0.40 (±0.23) |
| Air2stream (3-par) | 3.13 (±1.47) | 3.95 (±1.55) | 0.02 (±0.29) | 0.14 (±0.30) | -0.42 (±1.92) | 0.25 (±0.22) |
| Air2stream (4-par) | 3.15 (±1.29) | 4.01 (±1.40) | -0.14 (±1.01) | 0.14 (±0.29) | -0.49 (±1.77) | 0.24 (±0.23) |
| Air2stream (5-par) | 3.36 (±1.09) | 4.13 (±1.18) | -0.19 (±0.64) | 0.06 (±0.32) | -0.81 (±1.65) | 0.21 (±0.22) |
| Air2stream (7-par) | 3.85 (±1.23) | 4.81 (±1.45) | -0.80 (±1.41) | 0.01 (±0.30) | -1.27 (±2.15) | 0.15 (±0.20) |
| Air2stream (8-par) | 3.99 (±1.37) | 5.10 (±1.92) | -1.27 (±3.46) | -0.04 (±0.48) | -1.25 (±2.18) | 0.13 (±0.19) |
| MR | 3.55 (±2.00) | 4.42 (±2.22) | 0.13 (±0.35) | 0.13 (±0.36) | -0.28 (±2.61) | 0.20 (±0.23) |
| **Ensemble** | **2.09 (±0.86)** | **2.65 (±1.04)** | **0.31 (±0.78)** | **0.52 (±0.28)** | **-0.33 (±1.07)** | **0.55 (±0.18)** |

**Table A6: Evaluation of RF model performance during the training (raw datasets) and testing (raw datasets) phases considering the annual datasets. Mean MAE, RMSE, NSE, KGE, bias and R2 (with standard deviation) between observed and predicted WT values**

| Annual | TRAIN (raw datasets) | | | | | | TEST (raw datasets) | | | | | |
|---|---|---|---|---|---|---|---|---|---|---|---|---|
| Station/metric | MAE (ºC) | RMSE (ºC) | NSE | KGE | bias (ºC) | $R^2$ | MAE (ºC) | RMSE (ºC) | NSE | KGE | bias (ºC) | $R^2$ |
| 1 | 1.18 (±1.77) | 1.40 (±2.13) | 0.96 (±0.40) | 0.80 (±0.37) | 0.15 (±0.15) | 0.99 (±0.29) | 3.53 (±0.56) | 5.03 (±0.48) | -0.01 (±0.22) | 0.33 (±0.20) | 1.56 (±1.02) | 0.17 (±0.06) |
| 7 | 1.07 (±0.65) | 1.55 (±0.61) | 0.84 (±0.20) | 0.69 (0.19±) | -0.08 (±0.08) | 0.91 (±0.11) | 2.26 (±0.36) | 3.19 (±0.28) | 0.46 (0.11±) | 0.50 (±0.17) | 0.01 (±0.16) | 0.46 (±0.09) |
| 12 | 0.78 (±0.22) | 0.97 (±0.30) | 0.92 (±0.07) | 0.81 (±0.06) | 0.03 (±0.03) | 0.95 (±0.07) | 2.98 (±0.06) | 3.53 (±0.10) | 0.15 (±0.05) | 0.35 (±0.08) | -1.56 (±0.05) | 0.32 (±0.05) |
| 13 | 0.83 (±0.20) | 1.04 (±0.26) | 0.86 (0.09±) | 0.71 (±0.08) | 0.04 (±0.03) | 0.93 (±0.08) | 2.51 (±0.09) | 3.09 (±0.09) | 0.29 (±0.05) | 0.20 (±0.04) | 0.28 (±0.15) | 0.50 (±0.09) |
| 22 | 1.22 (±0.69) | 1.66 (±0.49) | 0.91 (±0.08) | 0.79 (±0.07) | -0.19 (±0.03) | 0.94 (±0.07) | 2.57 (±0.09) | 3.30 (±0.11) | 0.45 (±0.04) | 0.57 (±0.05) | -1.67 (±0.06) | 0.60 (±0.05) |
| 29 | 0.61 (±0.16) | 0.88 (±0.19) | 0.95 (±0.03) | 0.88 (±0.03) | -0.02 (±0.01) | 0.96 (±0.03) | 1.47 (±0.06) | 2.32 (±0.06) | 0.62 (0.02±) | 0.60 (±0.03) | -0.16 (±0.05) | 0.65 (±0.02) |
| 30 | 0.85 (±0.16) | 0.92 (±0.20) | 0.92 (±0.05) | 0.85 (±0.04) | 0.01 (±0.01) | 0.93 (±0.05) | 1.56 (±0.06) | 2.31 (±0.06) | 0.44 (±0.03) | 0.67 (±0.03) | -1.09 (±0.05) | 0.56 (±0.03) |
| 46 | 0.69 (±0.25) | 0.94 (±0.34) | 0.96 (±0.04) | 0.91 (±0.06) | 0.01 (±0.02) | 0.97 (±0.03) | 2.79 (±0.09) | 3.63 (±0.14) | 0.60 (±0.03) | 0.61 (±0.07) | 0.32 (±0.54) | 0.61 (±0.02) |
| 59 | 0.65 (±0.18) | 0.83 (±0.25) | 0.96 (±0.03) | 0.90 (±0.04) | 0.01 (±0.02) | 0.97 (±0.03) | 1.20 (±0.06) | 1.57 (±0.11) | 0.84 (±0.02) | 0.90 (±0.05) | 0.33 (±0.06) | 0.85 (±0.02) |
| 60 | 0.71 (±0.23) | 0.92 (±0.30) | 0.96 (±0.04) | 0.90 (±0.06) | -0.03 (±0.02) | 0.97 (±0.03) | 2.00 (±0.05) | 2.48 (±0.07) | 0.70 (±0.02) | 0.78 (±0.06) | 0.85 (±0.13) | 0.74 (±0.02) |
| 73 | 0.84 (±0.24) | 1.16 (±0.27) | 0.92 (±0.05) | 0.81 (±0.05) | 0.04 (±0.01) | 0.95 (±0.04) | 1.92 (±0.14) | 2.47 (±0.12) | 0.58 (±0.04) | 0.63 (±0.07) | 0.28 (±0.05) | 0.58 (±0.04) |
| 83 | 0.65 (±0.15) | 0.88 (±0.20) | 0.95 (±0.03) | 0.87 (±0.03) | 0.00 (±0.02) | 0.96 (±0.03) | 1.61 (±0.08) | 2.07 (±0.12) | 0.69 (±0.04) | 0.72 (±0.05) | -0.13 (±0.04) | 0.69 (±0.04) |
| Average | **0.84** (±0.21) | **1.09** (±0.28) | **0.93** (±0.04) | **0.83** (±0.07) | **0.00** (±0.08) | **0.95** (±0.02) | **2.20** (±0.70) | **2.92** (±0.92) | **0.48** (±0.24) | **0.57** (±0.20) | **-0.08** (±0.95) | **0.56** (±0.18) |

**Table A7: Evaluation of RF model performance during the training (modified datasets) and testing (raw datasets) phases considering the annual datasets. Mean MAE, RMSE, NSE, KGE, bias and $R^2$ (with standard deviation) between observed and predicted WT values**

| Annual | TRAIN (modified datasets) | | | | | | TEST (raw datasets) | | | | | |
|---|---|---|---|---|---|---|---|---|---|---|---|---|
| Station/metric | MAE (ºC) | RMSE (ºC) | NSE | KGE | bias (ºC) | $R^2$ | MAE (ºC) | RMSE (ºC) | NSE | KGE | bias (ºC) | $R^2$ |
| 1 | 2.99 (±1.24) | 3.20 (±1.47) | 0.77 (±0.31) | 0.77 (±0.30) | 0.00 (±0.15 | 0.77 (±0.15) | 3.44 (±0.34) | 5.00 (±0.31) | -0.04 (±0.14) | 0.37 (±0.22) | 1.26 (±1.28) | 0.17 (±0.07) |
| 7 | 0.99 (±0.45) | 2.35 (±0.66) | 0.49 (±0.09) | 0.58 (±0.12) | 0.00 (±0.07) | 0.49 (±0.08) | 1.86 (±0.43) | 2.62 (±0.38) | 0.64( ±0.14) | 0.65 (±0.12) | 0.16 (±0.62) | 0.65 (±0.11) |
| 12 | 0.00 (±0.38) | 0.00 (±0.57) | 1.00 (±0.08) | 1.00 (±0.11) | 0.00 (±0.05) | 1.00 (±0.07) | 2.54 (±0.62) | 3.25 (±0.78) | 0.28 (±0.50) | 0.23 (±0.21) | -0.54 (±0.98) | 0.37 (±0.12) |
| 13 | 0.98 (±0.33) | 1.31 (±0.45) | 0.79 (±0.13) | 0.75 (±0.15) | -0.04 (±0.05) | 0.80 (±0.11) | 1.78 (±0.50) | 2.55 (±0.54) | 0.52 (±0.29) | 0.51 (±0.34) | -0.86 (±1.08) | 0.63 (±0.19) |
| 22 | 0.09 (±0.69) | 0.17 (±0.89) | 1.00 (±0.11) | 0.99 (±0.15) | 0.00 (±0.07) | 1.00 (±0.09) | 2.47 (±0.78) | 3.23 (±0.90) | 0.47 (±0.43) | 0.54 (±0.15) | -1.66 (±1.57) | 0.65 (±0.15) |
| 29 | 0.47 (±0.36) | 0.72 (±0.54) | 0.96 (±0.06) | 0.97 (±0.08) | 0.00 (±0.05) | 0.96 (±0.05) | 1.36 (±0.41) | 1.81 (±0.52) | 0.77 (±0.21) | 0.85 (±0.13) | -0.25 (±1.00) | 0.77 (±0.13) |
| 30 | 0.33 (±0.29) | 0.57 (±0.47) | 0.98 (±0.07) | 0.98 (±0.09) | 0.00 (±0.03) | 0.98 (±0.06) | 1.47 (±0.47) | 2.24 (±0.50) | 0.47 (±0.32) | 0.75 (±0.11) | -0.43 (±0.71) | 0.57 (±0.09) |
| 46 | 1.11 (±0.40) | 1.68 (±0.58) | 0.93 (±0.05) | 0.90 (±0.06) | -0.09 (±0.05) | 0.94(±0.04) | 2.15 (±0.66) | 2.95 (±0.85) | 0.73 (±0.23) | 0.73 (±0.13) | -0.92 (±1.50) | 0.77 (±0.11) |
| 59 | 0.51 (±0.33) | 0.78 (±0.44) | 0.96 (±0.04) | 0.90 (±0.07) | -0.01 (±0.05) | 0.97 (±0.04) | 1.09 (±0.40) | 1.43 (±0.52) | 0.87 (±0.16) | 0.91 (±0.12) | 0.11 (±0.67) | 0.87 (±0.10) |
| 60 | 0.73 (±0.35) | 1.02 (±0.47) | 0.96 (±0.04) | 0.91 (±0.05) | -0.02 (±0.03) | 0.97 (±0.04) | 1.83 (±0.36) | 2.41 (±0.53) | 0.72 (±0.17) | 0.83 (±0.11) | 0.60 (±0.99) | 0.74 (±0.12) |
| 73 | 0.84 (±0.39) | 1.16 (±0.52) | 0.92 (±0.06) | 0.81 (±0.08) | 0.04 (±0.05) | 0.95 (±0.04) | 1.92 (±0.39) | 2.47 (±0.47) | 0.58 (±0.21) | 0.69 (±0.13) | 0.28 (±1.17) | 0.58 (±0.11) |
| 83 | 0.65 (±0.28) | 1.06 (±0.37) | 0.91 (±0.04) | 0.84 (±0.06) | 0.01 (±0.03) | 0.93 (±0.03) | 1.61 (±0.41) | 2.05 (±0.49) | 0.69 (±0.21) | 0.73 (±0.14) | -0.34 (±0.72) | 0.70 (±0.10) |
| **Average** | 0.81 (±0.77) | 1.17 (±0.90) | 0.89 (±0.15) | 0.87 (±0.12) | -0.01 (±0.03) | 0.90 (±0.15) | 1.96 (±0.63) | 2.67 (±0.91) | 0.56 (±0.25) | 0.65 (±0.20) | -0.22 (±0.77) | 0.62 (±0.19) |

**Table A8: SMOGN parameters for the best RF predictions**

| Station | k | samp_method | rel_thres | rel_coef | rel_xtrm_type |
|---|---|---|---|---|---|
| 1 | 4.0 | extreme | 0.53 | 0.02 | high |
| 7 | 3.0 | extreme | 0.46 | 0.36 | both |
| 12 | 2.0 | extreme | 0.29 | 0.17 | both |
| 13 | 7.0 | extreme | 0.81 | 0.02 | high |
| 22 | 3.0 | balance | 0.46 | 0.01 | both |
| 29 | 7.0 | extreme | 0.63 | 0.14 | high |
| 30 | 6.0 | extreme | 0.98 | 0.40 | both |
| 46 | 2.0 | balance | 0.62 | 0.29 | both |
| 59 | 5.0 | extreme | 0.40 | 0.10 | both |
| 60 | 5.0 | extreme | 0.77 | 0.17 | both |
| 73 | 5.0 | extreme | 0.53 | 0.38 | both |
| 83 | 5.0 | extreme | 0.04 | 0.28 | both |

**Table A9: Evaluation of the Random Forest performance during the training and testing phases considering the annual datasets and the sequential increase of the models' predictors. Mean MAE, RMSE, NSE, KGE, bias and $R^2$ (with standard deviation) between observed and predicted WT values. 1) mean air temperature; 2) mean air temperature + discharge; 3) mean air temperature + discharge + radiation; 4) mean air temperature + discharge + radiation + maximum air temperature; 5) mean air temperature + discharge + radiation + maximum air temperature + minimum air temperature; 6) mean air temperature + discharge + radiation + maximum air temperature + minimum air temperature + MOY; 7) mean air temperature + discharge + radiation + maximum air temperature + minimum air temperature + MOY + DOY.**

| Annual | TRAIN | | | | | | |
|---|---|---|---|---|---|---|---|
| Metric/predictor | 1) | 2) | 3) | 4) | 5) | 6) | 7) |
| MAE (ºC) | 1.84 (±0.51) | 1.61 (±0.45) | 1.50 (±0.41) | 1.48 (±0.40) | 1.44 (±0.40) | 1.41 (±0.40) | 1.07 (±0.30) |
| RMSE (ºC) | 2.35 (±0.57) | 2.09 (±0.51) | 1.98 (±0.47) | 1.94 (±0.47) | 1.90 (±0.46) | 1.86 (±0.46) | 1.43 (±0.38) |
| NSE | 0.72 (±0.10) | 0.78 (±0.09) | 0.80 (±0.07) | 0.81 (±0.07) | 0.82 (±.07) | 0.82 (±0.07) | 0.89 (±0.06) |
| KGE | 0.67 (±0.12) | 0.70 (±0.11) | 0.71 (±0.11) | 0.71 (±0.10) | 0.72 (±0.10) | 0.72 (±0.10) | 0.82 (±0.09) |
| bias (ºC) | 0.00 (±0.11) | 0.01 (±0.09) | 0.01 (±0.08) | 0.01 (±0.09) | 0.01 (±0.11) | 0.00 (±0.08) | 0.00 (±0.08) |
| $R^2$ | 0.76 (±0.08) | 0.82 (±0.07) | 0.85 (±0.05) | 0.86 (±0.04) | 0.87 (±0.04) | 0.87 (±0.05) | 0.92 (±0.04) |
| Annual | TEST | | | | | | |
| Metric/predictor | 1) | 2) | 3) | 4) | 5) | 6) | 7) |
| MAE (ºC) | 3.55 (±0.97) | 3.43 (±1.01) | 3.37 (±1.10) | 3.35 (±1.08) | 3.35 (±1.10) | 3.29 (±1.10) | 2.51 (±0.95) |
| RMSE (ºC) | 4.54 (±1.18) | 4.40 (±1.17) | 4.30 (±1.19) | 4.29 (±1.19) | 4.30 (±1.23) | 4.23 (±1.21) | 3.29 (±1.12) |
| NSE | 0.03 (±0.35) | 0.08 (±0.34) | 0.12 (±0.35) | 0.13 (±0.34) | 0.13 (±0.34) | 0.16 (±0.33) | 0.48 (±0.26) |
| KGE | 0.32 (±0.25) | 0.32 (±0.26) | 0.31 (±0.28) | 0.32 (±0.27) | 0.31 (±0.28) | 0.33 (±0.28) | 0.60 (±0.20) |
| bias (ºC) | -0.15 (±1.33) | -0.26 (±1.37) | -0.25 (±1.18) | -0.22 (±1.21) | -0.22 (±1.26) | -0.21 (±1.21) | -0.10 (±1.25) |
| $R^2$ | 0.23 (±0.20) | 0.26 (±0.21) | 0.28 (±0.22) | 0.28 (±0.23) | 0.28 (±0.23) | 0.29 (±0.23) | 0.58 (±0.18) |

**Code and data availability**. The python code used to generate all results for this publication and the Fortran code of the Air2stream model can be found in Almeida and Coelho (2022). Additionally, this repository includes the input data considered in this study (83 datasets). It is also possible to download the code/data from https://github.com/mcvta/WaterPythonTemp.

**Author contributions.** MA conceived the study, performed the simulations and wrote the manuscript. PC, contributed to the study design and to the results analysis. All authors contributed to the discussion and manuscript revision. All authors read and approved the final manuscript.

**Competing interests.** The authors declare that they have no conflict of interest.

**Acknowledgements.** This study had the support of national funds through Fundação para a Ciência e Tecnologia (FCT), under the project LA/P/0069/2020 granted to the Associate Laboratory ARNET, and the strategic project UDIB/04292/2020 granted to MARE.

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
