# Peer review of "Modeling river water temperature with limiting forcing data: air2stream v1.0.0, machine learning and multiple regression"

_Geoscientific Model Development, 2022_

## Author Comment (AC1)

Referee 1: Determining inflow temperatures of rivers or streams into receiving water bodies is a critical need. The authors are to be commended for addressing this topic. The overall theme of the paper was not crystal clear – was this about modeling river water temperatures themselves or modeling the inflows into receiving water models? Did the modeling include hydrology models predicting depth and flow? If so, then it was not clear that the models used were compared to flow and depth data which are critical for modeling river temperatures. Also, the overall RMSE error for the models was much higher than accepted river temperature models. This leads to the conclusion that there were underlying issues in the datasets used in the model. If the datasets were improved, would the conclusions of the study have been any different? An important aspect of this paper that was not evaluated was the advantages and disadvantages of using a physical based model rather than correlations and regressions to model water temperature.

RESPONSE: Thank you for the time spent and for the thoughtful comments and suggestions towards improving our manuscript. To facilitate the work of the reviewers and editor, we refer to the former manuscript indicating the line that was modified.

Referee 1: "The overall theme of the paper was not crystal clear – was this about modeling river water temperatures themselves or modeling the inflows into receiving water models?"

RESPONSE: Thank you for pointing this out. The methodological approach considered in this study was defined to improve the thermal characterization of the lake/reservoir water quality models' boundary condition. This was the reason for undertaking the study - a practical need to improve the water temperature (WT) characterization at these sections. We assume that there are no significant variations between the water quality station observed WT, and the WT at the downstream portion of a river, which normally coincides with the lake/reservoir water quality model boundary condition. We could have just stated that we wanted to simulate the water temperature of rivers, however, in our opinion, it was important to reflect about the rationale behind the objective of this study regardless of the assumptions made.

To clarify the reviewer's concern the following sentence was included in the abstract:

Page 1 – Line 11: "Commonly, the WT observed in monitoring stations located near the downstream section of rivers are assumed to be the boundary condition of lake/reservoir water quality models. The main goal of this study is to identify a suitable WT modeling solution for these sections given the scarcity of the forcing datasets."

The following sentences were included in the introduction section:

Page 3 – Line 91: "Hence, the main objective of this study is to identify a suitable WT modeling solution to improve the lake/reservoir water quality models' boundary condition. It is important to mention that, for this study, an absence of significant variation between the water quality station observed WT and the WT at the downstream portion of a river was assumed, which coincides with the lake/reservoir water quality model boundary condition."

Page 4 – Line 97 – "This modeling solution will be considered to improve the characterization of lake/reservoir WT boundary conditions (assuming that the observed WT of the downstream sections of the rivers are the boundary condition of lake/reservoir water quality models)."

The following sentence was also included in the methodology definition section:

Page 7 – Line 151: "iv) There are no significant variations between the water quality station observed WT, and the WT at the downstream portion of a river, which coincides with the lake/reservoir water quality model boundary condition."

Referee 1: "Did the modeling include hydrology models predicting depth and flow? If so, then it was not clear that the models used were compared to flow and depth data which are critical for modeling river temperatures."

RESPONSE: No, the modeling approach didn't include hydrology models predicting depth and flow. This fact was clarified by the inclusion of a table with the models' input/output (Page 7 – Line 163 – Table 3). A reservoir can have a high number of tributaries and the application of a physical based model is commonly limited by the availability of data required, not only for the river's conceptual representation but also for the calibration of the model (e.g., observed depth time series). It is also relevant to mention that, as shown by Toffolon and Piccolroaz (2015), the river's thermal response on a daily timescale does not strongly depend on the flow depth.

Referee 1: "Also, the overall RMSE error for the models was much higher than accepted river temperature models. This leads to the conclusion that there were underlying issues in the datasets used in the model."

RESPONSE:

The quantity and quality of the training and testing datasets was one of the primary reasons for undertaking this study. In practice these are the datasets that will be considered for the characterization of the boundary conditions of lake/reservoir water quality models for this region. The Air2stream model and the multiple

regression approach were included in this study to define a benchmark. It is well established that both approaches perform well with regard to predicting water temperature values. Therefore, in our opinion the results obtained with these models should be considered as the reference point for the analysis of the ML (machine learning) algorithm performance. Additionally, the results of this study should be compared with studies that also have a significant amount of missing daily river water temperature values. Piccolroaz (2016), modeled the water temperature of two lakes located in the USA (Lake Erie and Lake Superior) and showed that when the length of the calibration period is one year and the percentage of missing data is in the range of 99%, the RMSE between observed and predicted lake water temperature is >3.5°C.

We agree with the reviewer, it is common to obtain RMSE values in river water temperature modeling studies of less than 1°C. In fact, the manuscript includes a list of reviewed publications on river water temperature modeling and the corresponding RMSE between observed and modelled water temperature values. It is important to mention that most of these simulations include well characterized rivers in relation to the forcing meteorology and water temperature observed values. Furthermore, they are limited to a small number of rivers. The RMSE values varied from 0.42°C (Feigl et al., 2021) to 2.74°C (Zhu et al., 2019). This last value is high and similar to the overall mean RMSE error obtained considering the models ensemble results of this study of 2.75°C.

We also agree that there are underlying issues in the datasets, namely:

i)   The significant amount of daily river WT values that are missing (96.9% to 99.9%) ($\mu$= 98.8%; $\sigma$=0.68) (e.g., the minimum training dataset as 11 water temperature values). In such small datasets an outlier can have a significant effect on the RMSE, and the models' overfitting will represent an important drawback;

ii)  The length of the simulation period: 1980-2020. The training/testing dataset spans a long period of time. Hence, the interannual variation of river entering fluxes (e.g. hydropower release) may have a significant effect on the quality of the training and testing dataset;

iii) The fact that the meteorological forcing was obtained from ERA5 reanalysis. McNicholl et al. (2021) found large biases between the satellite and land data for a temperate region (Dublin) and for a tropical region (Singapore). Furthermore, the revised version of the manuscript will include the comparison of eleven ERA5 daily air temperature values with observed air temperature values (we have considered all meteorological stations located within a 5 km radius). Results show that the annual RMSE obtained between the two datasets considering all stations varied from 0.02°C to 4.15°C ($\mu$=1.43°C; $\sigma$=0.55°C ).  This interannual variability was also found in the water temperature modeling results, which suggests that the

meteorological forcing also has a significant impact on the overall RMSE obtained in this study;

iv) The effect of upstream conditions.

The outliers of the datasets could have been removed and synthetic samples could have been generated for some poorly represented ranges. The performance of the models would be significantly increased. However, in our opinion, this process must be driven by the lake/reservoir water quality model calibration.

We have included the following paragraph to clarify the reviewer's concern regarding the quality of the datasets:

Page 29 – Line 558: "It is also relevant to mention that the results of this study suggest that, besides the WT dataset gaps, the modeling results were also affected by the presence of a large number of WT outliers, by the uncertainty induced by the mean air temperature ERA5 reanalysis datasets and by upstream conditions, which increases with the watershed area. The results of this study considering the quality of the input datasets suggests that when the missing datasets reach 98%, a RMSE <3.0ºC is indicative of a good modeling performance. Also relevant is the fact that this error can be further decreased by the generation of synthetic samples to some poorly represented ranges within the datasets, by applying a model such as SMOGN (Branco et al. 2017)."

Referee 1: "If the datasets were improved, would the conclusions of the study have been any different?"

RESPONSE: The main conclusion of this study would be the same: from a practical modeling perspective, when the number of predictor variables and observed river WT values are limited, the application of all the models considered in this study, considering the hyperparameter optimization algorithm (hyperopt) is quite relevant. The machine learning model results obtained in this study are very similar, hence, the best model, Random Forest, can be replaced by the SVR or by the ANN.

It is also important to mention that by improving the datasets we assume that the reviewer means removing outliers and adding synthetic samples to the datasets by applying a model, such as SMOGN. A pre-processing approach for imbalanced regression tasks (Branco et al., 2017). In fact, the modeling results of this study could have been significantly improved with the implementation of a pre-processing approach, such as SMOGN. This algorithm was not implemented because the user needs to assign a greater degree of importance to the predictive performance obtained for some poorly represented ranges compared to other more frequent

ranges. In our opinion, this process needs to be driven by the water quality model temperature calibration process. Hence, we have chosen to preserve the original datasets and to evaluate the model's performance over the raw datasets.

Referee 1: "An important aspect of this paper that was not evaluated was the advantages and disadvantages of using a physical based model rather than correlations and regressions to model water temperature."

RESPONSE: We agree with the reviewer. It would have been interesting to evaluate the advantages and disadvantages of using a physical based model. However, as previously mentioned, the application of a physical based model to 83 rivers is limited due to the significant amount of data required, including stream geometry, land use, meteorological conditions, and heat flux components, which are difficult to compute. However, the analysis does include a hybrid model characterized by a physical based structure associated with a stochastic calibration of the model parameters, Air2stream v1.0.0. (Toffolon and Piccolroaz, 2015) considering five different parametrizations. The results of Air2stream and multiple regression were included to define a benchmark for the overall modeling results.

**Specific comments:**

Referee 1: "Abstract: Line 16: define variables used for 'Multiple Regression' – this should be a separate sentence in the Abstract where the variables used for the different approaches are described."

RESPONSE: We agree with the reviewer. The following sentence was included:

Page 1 – Line 14: "With the exception of Air2stream, which was forced with mean daily air temperature and discharge, all other models were forced with: mean, maximum, and minimum daily air temperature, mean daily total radiation (shortwave), mean daily discharge, month of the year and day of the year."

Referee 1: "After reading the abstract, the reader is not left with a better understanding on how to fill in data gaps, other than just take more measurements!"

RESPONSE: To clarify this point the following sentence was included:

Page 1 – Line 20: "Therefore, the datasets gaps can be filled with the best model of the ensemble approach."

Referee 1: "Line 44-45: 'common practice to average out sub-daily effects and to consider a daily discretization for modeling purposes' – the impacts of this should be explored further since this can impact significantly the waterbody being modeled"

RESPONSE: We agree with the reviewer. This is a very important and complex challenge for water quality modelers: increase the sub-daily samples of flow and water temperature values for the cases when only a dataset of mean daily inflow values is available. We have included the following sentence to stress the importance of this limitation:

Page 2 – Line 46: "This assumption can have a significant impact on lake/reservoir water quality modeling results, namely when lake/reservoir inflows are large. The fall and spring turnover onset, stratification strength/length and the overall heat budget can be affected."

Referee 1: "Line 45: 'Air temperature approximates the equilibrium temperature of a river and is, therefore, frequently used as the independent variable;' – these are indeed different even though the air temperature responds to the same atmospheric forcing as the equilibrium temperature of the waterbody. So, the text should read that air and equilibrium temperature correlate – but are not approximations for each other."

RESPONSE: We agree with the reviewer. Thank you for pointing this out. This sentence was changed:

Page 2 – Line 46: "Air temperature correlates with the equilibrium temperature of a river and is, therefore..."

Referee 1: "Line 65-70: Almost as important as the approach used are the variables used in the regression and other models. Could a listing of the predictor variables be itemized? Or is it just air temperature?"

RESPONSE: We agree with the reviewer. The following table was included on the revised version of the manuscript:

Page 7 – Line 163:

Table 3: Model predictor variables

| Model | Predictor variables | Output variable |
|---|---|---|
| RF | Mean, max., and min. daily air temperature (ºC) Mean daily total radiation (shortwave) ($Jm^{-2}$) Mean daily discharge ($m^3.s^{-1}$) MOY and DOY | Water temperature |
| ANN | Mean, max., and min. daily air temperature (ºC) Mean daily total radiation (shortwave) ($Jm^{-2}$) | |

| | |
|---|---|
| | Mean daily discharge ($m^3.s^{-1}$)
 MOY and DOY |
| **SVR** | Mean, max., and min. daily air temperature (ºC)
 Mean daily total radiation (shortwave) ($Jm^{-2}$)
 Mean daily discharge ($m^3.s^{-1}$)
 MOY and DOY |
| **Air2stream** | Mean daily air temperature (ºC)
 Mean daily discharge ($m^3.s^{-1}$) |
| **ML** | Mean, max., and min. daily air temperature (ºC)
 Mean daily total radiation (shortwave) ($Jm^{-2}$)
 Discharge ($m^3.s^{-1}$)
 MOY and DOY |

Referee 1: "Line 74: Why single out wind velocity? One also needs air temperature, dew point temperature (or relative humidity), cloud cover and short-wave solar radiation also to compute the heat balance."

RESPONSE: Because it is very difficult to have accurate shading and daily wind velocity and direction datasets for unmonitored (in regard to meteorology) water quality stations. However, we think that the sentence will be improved with the inclusion of the reviewer's suggestion.

Page 3 – Line 74: "…although they do require a large amount of forcing data, including stream geometry, air temperature, dew point temperature (or relative humidity), cloud cover and short-wave solar radiation, degree of shading and wind direction/ velocity."

Referee 1: "Line 76-77: It is not clear if the river modeling is using the predictors or if it is just the boundary conditions are being predicted for use in a river model."

RESPONSE: The reviewer is right. This sentence is not clear and was changed:

Page 3 – Line 77: "The number and type of predictor variables considered to force river WT models in several intercomparison studies is quite different."

Referee 1: "Table 1: I assume the error statistics are for the river WT – not the boundary conditions. So, I assumed in reading this that for each of these models, there was no explicit river model other than the correlations/stochastic models. But as a reader I am confused since I thought the intent of the paper was focused on boundary conditions and techniques to determine boundary conditions."

RESPONSE: The reviewer is right, the error statistics are for river WT, and not for the boundary conditions. We think that the sentences included in the abstract and on pages 3, 4 and 7 clarifies the main goal of the study.

Referee 1: "Line 86/87/105: Now the focus moves to lake or reservoir models. I thought the focus was on river models as the receiving water body – see Line 96. I agree that developing the boundary conditions would benefit lake/reservoir models, but the focus in the abstract and throughout the paper needs to be refocused to include any receiving water quality model, not just rivers."

RESPONSE: The reviewer is right. The error statistics are for river WT and not for the boundary conditions. Also in this case, we think that the sentences included in the abstract and on pages 3, 4 and 7 clarifies the main goal of the study.

Referee 1: "Line 136/Table 2: I have no idea what the total number refers to in Table 2 nor the statistics. Are these 'predictors' or are these WT in the rivers? What are the units of mean, etc.? And please itemize clearly what the predictors are for training and validation. Or do they vary? This is critical to understanding if this approach can be used by others."

RESPONSE: The total number shown in Table 2 is the sum of all WT values considering all stations. The predictors are the same for all models except for the Air2stream model. The inclusion of Table 3, in our opinion, clarifies this point.

Referee 1: "Line 143: 'model a significant number of watersheds' – does this mean just for WT and flow? Or does it include stage also?"

RESPONSE: Only WT was modeled. We think that the inclusion of Table 3 clarifies this point as it shows that the output variable for all models is WT.

Referee 1: "Line 148/149: Not having on-site meteorological data is large weakness of this study. We have found that the on-line estimates are often poor and significantly affect the model predictions. In the basin where you did your analysis, there must be some meteorological stations that could have been used for ground-truthing the ERA5 'data'. Doing that comparison would also help inform readers of the bias in using estimates when there is no on-site meteorological data."

RESPONSE: We agree with the reviewer. A new section has been included in the manuscript to describe the evaluation of the mean daily ERA5 air temperature values

considering all meteorological stations located within a 5 km radius of the studied stations (the following text, table and figure were included):

Page 7 – Line 166: The results section starts with the evaluation of the ERA5 mean daily air temperature datasets. These datasets were compared with ground measurements of mean daily air temperature considering all the meteorological datasets located within a 5 km radius of the stations considered in this study."

Page 13 – Line 326: "4.1 Air temperature - ERA5 versus ground observed datasets

In this analysis the observed air temperature datasets of a total of eleven meteorological stations were considered. These are all available air temperature datasets observed within a 5 km radius of the stations considered in this study. Results show that the mean RMSE obtained between the two datasets considering all stations varied from 1.26ºC to 2.05ºC (μ=1.54ºC; σ=0.24ºC) and that, according to the mean bias values, the ERA5 tends to overestimate the observed air temperature datasets at 91% of the stations. Overall, a mean value of 1.54ºC (σ=0.24ºC) and a mean NSE value of 0.90 (σ=0.07) is indicative of a good performance. However, as shown in Figure 2, there are some sporadic significant discrepancies between the two datasets (Fig. 2). Additionally, results show that the stations with a RMSE higher than 2ºC are scattered all over the country. In this context it is relevant to mention that McNicholl et al. (2021) also found large biases between the ERA5 daily air temperature datasets and land data for a temperate region (Dublin) and for a tropical region (Singapore). Generally, these results suggest that the consideration of the ERA5 air temperature datasets for WT modeling can, sporadically, induce some important discrepancies between the two datasets. The error can significantly increase if the model's training/testing dataset is small.

**Table 5: Evaluation of ERA5 daily air temperature datasets - MAE, RMSE, NSE, KGE, bias and $R^2$ (with standard deviation) between observed and ERA5 values**

| Station | N* | MAE | RMSE | NSE | KGE | bias | $R^2$ |
|---|---|---|---|---|---|---|---|
| st4 | 80 | 1.10±0.26 | 1.39±0.28 | 0.91±0.04 | 0.94±0.05 | 0.74±0.47 | 0.94±0.02 |
| st6 | 120 | 1.10±0.37 | 1.34±0.38 | 0.90±0.17 | 0.92±0.09 | -0.15±0.79 | 0.90±0.11 |
| st30 | 98 | 1.31±0.29 | 1.72±0.40 | 0.91±0.07 | 0.95±0.06 | -0.48±0.70 | 0.92±0.05 |
| st32 | 67 | 1.16±0.52 | 1.43±0.58 | 0.96±0.04 | 0.94±0.05 | -0.75±0.90 | 0.97±0.02 |
| st38 | 110 | 0.88±0.34 | 1.26±0.48 | 0.94±0.09 | 0.96±0.06 | -0.46±0.57 | 0.95±0.04 |
| st42 | 21 | 1.19±0.47 | 1.53±0.58 | 0.93±45.49 | 0.87±2.22 | -0.42±0.75 | 0.94±0.03 |
| st50 | 90 | 1.08±0.30 | 1.45±0.48 | 0.91±0.06 | 0.89±0.11 | -0.14±0.39 | 0.92±0.04 |
| st62 | 24 | 1.30±0.74 | 1.67±0.80 | 0.89±6.28 | 0.94±0.92 | -0.17±1.36 | 0.90±0.04 |
| st68 | 47 | 1.60±1.18 | 2.05±1.09 | 0.71±9.81 | 0.86±0.23 | -1.47±1.24 | 0.88±0.2 |
| st83 | 137 | 1.49±0.40 | 1.79±0.39 | 0.92±0.03 | 0.94±0.04 | -0.60±0.88 | 0.93±0.02 |
| st91 | 51 | 1.04±0.13 | 1.33±0.16 | 0.93±0.04 | 0.96±0.09 | -0.46±0.47 | 0.94±0.04 |

*number of dataset values

[Figure]

**Figure 2: Metrics histograms of daily air temperature - ERA5 *versus* ground observed datasets**

Additionally, the following sentence was included in the discussion section:

Page 13 – Line 558: "It is also relevant to mention that the results of this study suggest that, besides the WT dataset gaps, the modeling results were also affected by the presence of a large number of WT outliers, by the uncertainty induced by the mean air temperature ERA5 reanalysis datasets and by upstream conditions, which increases with the watershed area. The results of this study considering the quality of the input datasets suggests that when the missing datasets reach 98%, a RMSE <3.0 °C is indicative of a good modeling performance.  Also relevant is the fact that this error can be further decreased by the generation of synthetic samples to some poorly represented ranges within the datasets, by applying a model such as SMOGN (Branco et al. 2017)."

Referee 1: "Line 155: Why was this lapse rate chosen, -6.5oK/km? What are the impacts of assuming a fixed lapse rate over all your model domains?"

RESPONSE: Thank you for pointing this out. Initially this correction -6.0ºC /km (Faher and Harris, 2004) was applied to allow the comparison of the ERA5 air temperature reanalysis with ground measurements. The observed lapse rate is variable as a

function of location and other variables, however, in our opinion, this correction is relevant. A constant lapse rate does not affect the modeling results.

The following sentence was corrected:

Page 7 – Line 156:  … by considering a linear variation of air temperature with the altitude, dT/dz=-6.0°C/km.

Referee 1: "Line 163: Why is there a larger training dataset than a validation dataset? One would expect then with more training data that the results should be 'better trained' or more valid? What happens if the training and validation datasets were 80-20%?"

RESPONSE: If we are certain that the model correctly describes the process that we are modeling, then to determine the model parameters we can use all available data. However, in practice we are not sure that the model describes the phenomenon correctly. If we apply the model to the entire dataset the model will overfit the data and the algorithm, unfortunately, will not perform accurately against unseen data.

Hence, to avoid overfitting we divide the observations into training and testing datasets. Empirical analysis has shown that the best results are obtained if we consider 30%-20% for testing and the remaining percentage for training 70%-80% (Gholamy et al., 2018). Nguyen et al. (2021) shows that machine learning models are greatly affected by the training/testing ratios and concluded that the 30/70 ratio presented the best performance of the models. In this study we have chosen the 30/70 ratio because we have very small datasets for some stations and the consideration of a 20/80 ratio would determine a very small testing dataset.

Referee 1: "Line 196: 'The results from the various models were evaluated with six metrics considering the observed and predicted annual, dry and wet season datasets for river WT.' Does this imply that the predicted annual WT was used as a metric? Of what value is such a metric? Most river WT models are focused on maximum daily temperatures for fish habitat."

RESPONSE: Thank you for pointing this out. No, the models were used to predict daily values of WT. This sentence was rewritten:

Page 8 – Line 197:  The results from the various models were evaluated with six metrics considering the observed and predicted daily datasets of river WT. During the results evaluation three types of datasets were considered:

Annual datasets: All available daily averages of WT are compared to field data;

Wet season: Only the daily averages of WT corresponding to the wet season are compared to field data (October to March)

Dry season: Only the daily averages of WT corresponding to the dry season are compared to field data (April to September).

Referee 1: "Line 243 Eq (1): The equation has an error in the last term."

RESPONSE: Thank you for pointing this out. The equation was corrected.

Referee 1: "Eq (1): This river equation assumes that the flow rate is based on steady-state flow with no dispersion. I assume this model runs on a daily time step – assuming a new steady-state distribution each day? This should be clarified. Also, the term H is a critical parameter in this model, how was it determined and what were the meteorological variables necessary for its computation?"

RESPONSE: No, the model converges to a single steady state distribution. The model considers a dimensionless discharge (theta=(discharge/mean discharge)^1/m) where m is related with the exponent of the rating curve. The model is fitted to the entire input dataset (air temperature, water temperature and discharge) and the value of m and the value of all the other model coefficients are estimated during the model optimization process (calibration).

The following sentence was included:

Page 10 – Line 251: "The parameter, a4 is related with the exponent of the rating curve. The model is fitted to the entire input dataset (air temperature, water temperature and discharge) and the value of a4 and the value of all other model parameters are estimated during the model optimization process (calibration phase)."

In fact, H is the net heat flux at the water-atmosphere interface. The model assumes that air temperature can be used as a proxy for all surface heat fluxes. A Taylor series expansion is used to include the overall effect of air temperature (Toffolon and Piccolroaz, 2015).

The following sentence was included to clarify the reviewer's concern:

Page 10 – Line 247: "The model assumes that air temperature can be used as a proxy for all surface heat fluxes. A Taylor series expansion is used to include the overall effect of air temperature."

Referee 1: "Eq (4): What precisely were the regression variables used in this model?"

RESPONSE: The model predictor variables were included in Table 3: Mean, max., and min. daily air temperature (ºC), mean daily total radiation (shortwave) ($Jm^{-2}$) discharge ($m^3.s^{-1}$), month of the year (MOY) and day of the year (DOY).

Referee 1: "Line 285: Was time of concentration only for the hydrology prediction? What is the hydrology model prediction equation?"

RESPONSE: No, the watershed time of concentration was estimated because it encapsulates some of the main watershed characteristics that affect the river water temperature. The main objective was to evaluate whether the modeling error was correlated with this variable.

Referee 1: "Line 317: With the best performing model to have a RMSE of over 3oC is not convincing. For river temperature models, this type of error in the river or in the boundary conditions is too high. There must be other issues with your approach that lends itself to such a poor predictor. In our experience, river models (and lakes and reservoirs) are often well below 1oC RMSE. I would not use any of these approaches if it had such a high RMSE. And if you fixed the underlying issues, the best approach may change."

RESPONSE: In our opinion, the RMSE of the best performing value must be evaluated in the context of the study premises: 98% of missing data. This RMSE encapsulates the results of modeling a very large number of river sections (83) considering the datasets of a global climate reanalysis.

As water quality modelers we have very good results for lakes/reservoir and rivers WT predictions (RMSE < 1ºC ), considering ML solutions and physical-based models, such as QUAL2E or CE-QUAL-W2. But we also have poor results. This balance depends primarily on the quantity and quality of the available datasets.

Referee 1: "Table 4 – provide units. By annual datasets – what does that mean? You are comparing annual averages or daily averages to field data?"

RESPONSE: Thank you for pointing this out. The units were included. We are comparing daily averages to field data, considering the entire dataset which includes dry and wet season.

The following paragraph was included to clarify this point:

Page 10 – Line 197 "The results from the various models were evaluated with six metrics considering the observed and predicted daily datasets of river WT considering the observed and predicted annual, dry- and wet season datasets for river WT. During the results evaluation three types of datasets are considered:

Annual datasets: All available daily averages of WT are compared to field data,

Wet season: Only the daily averages of WT corresponding to the wet season are compared to field data (October to March),

Dry season: Only the daily averages of WT corresponding to the dry season are compared to field data (April to September)."

Referee 1: "Table 5 – provide units. Explain dry season datasets – daily data during dry season, or averaged data over an entire season?"

RESPONSE: Thank you for pointing this out. The units were included. We are comparing daily averages to field data obtained for the dry and wet season.

Table 7 – provide units.

RESPONSE: Thank you for pointing this out. The units were included.

**References**

Branco, P., Ribeiro, R. P., Torgo, L., Krawczyk, B., Moniz, N.: Smogn: a pre-processing approach for imbalanced regression, Proceedings of Machine Learning Research 74, 36–50. 2017.

Fahrer, C., and Harris, D., 2004. LAMPPOST A Mnemonic device for teaching climate variables. Journal of Geography 103:86-90, https://doi.org/10.1080/00221340408978579, 2004.

Feigl, M., Lebiedzinski, K., Herrnegger, M., and Schulz, K.: Machine-learning methods for stream water temperature prediction, Hydrol. Earth Syst. Sci., 25, 2951–2977, https://doi.org/10.5194/hess-25-2951-2021, 2021.

Gholamy, A., Kreinovich, V., Kosheleva, O.: Why 70/30 or 80/20 Relation Between Training and Testing Sets: A Pedagogical Explanation. Departmental Technical Reports (CS). 1209. https://scholarworks.utep.edu/cs_techrep/1209, 2018.

McNicholl, B., Lee, Y.H., Campbell, A.G., Dev, S.: Evaluating the Reliability of Air Temperature from ERA5 Reanalysis Data. IEEE Geosci. Remote Sens. Lett, 19, 1–5, https://doi.org/10.1109/lgrs.2021.3137643, 2021.

Nguyen, Q.H., Ly, H.-B., Ho, L.S., Al-Ansari, N., Van Le, H., Tran, V.Q., Prakash, I., Pham, B.T.: Influence of Data Splitting on Performance of Machine Learning Models in Prediction of Shear Strength of Soil. Math. Probl. Eng. 2021, 1–15, https://doi.org/10.1155/2021/4832864, 2021.

Piccolroaz, S.: Prediction of lake surface temperature using air 2stream model: guidelines, challenges, and future perspectives, Advances in Oceanography and Limnology, 7(1), 36-50, https://doi.org/10.4081/aiol.2016.5791, 2016.

Toffolon, M., and Piccolroaz, S.: A hybrid model for river water temperature as a function of air temperature and discharge, Environmental Research Letters, 10, 114011, https://doi.org/10.1088/1748-9326/10/11/114011, 2015.

Zhu, S., Hadzima-Nyarko, M., Gao, A., Wang, F., Wu, J., and Wu, S.: Two hybrid data-driven models for modeling water-air temperature relationship in rivers, Environ. Sci. Pollut. R., 26, 12622–12630, https://doi.org/10.1007/s11356-019-04716y, 2019.

---

## Author Comment (AC2)

Referee 2: "The authors conducted a study to explore the best method for predicting river water temperature in the case of limited observations. They compared the performance of five different methods, i.e., RF, ANN, SVR, the hybrid Air2stream model, Multiple Regression. In general, they found the ML techniques have better prediction performance with appropriate hyperparameters. RF achieved best evaluation metrics in their study. Based on the RF model, they studied the importance of input variables and the connection between the watershed time of concentration and model performance. However, I doubt the novelty of the study, since all the methods used in this study have been employed in previous studies in predicting river water temperature. Also, some of them have compared the performance of some of the methods, e.g., Zhu et al. (2018), Rajesh and Rehana (2021) and Rehana (2019). Moreover, the overall presentation is not clear to me. My comments are listed below."

RESPONSE: We sincerely appreciate all your valuable comments and suggestions, which helped us improve the quality of the manuscript. To facilitate the work of the reviewers and editor, we refer to the former manuscript indicating the line that was modified.

Referee 2: "I doubt the novelty of the study, since all the methods used in this study have been employed in previous studies in predicting river water temperature. Also, some of them have compared the performance of some of the methods, e.g.., Zhu et al. (2018), Rajesh and Rehana (2021) and Rehana (2019)"

RESPONSE: Thank you for pointing this out. We agree with the reviewer. The ML algorithms considered in this study have been previously considered to model river water temperature. Please note that this fact is shown in Table 1 where we present a list of reviewed publications on river WT modelling. In these studies, the number of sites modeled is quite small with one exception: the study conducted by DeWeber and Wagner (2013) which considered an ANN. The studies conducted by Moore et al. (2003) and Ducharne (2008) applied a multiple regression model. Additionally, the studies were conducted using large observed air temperature and water temperature datasets. Even so, the predicted RMSE considering all study results varies from 0.42°C to 2.74°C (μ=1.15°C; σ=0.6°C). This variability is mainly caused by:

i) water temperature measurement errors (e.g., sampling depth variability);
ii) the fact that river WT is not only affected by local environmental conditions but also by upstream conditions (Moore et al., 2005);
iii) watershed morphological differences across the different regions;
iv) differences in the type of models applied and in the model parameterization;
v) the consideration of different model predictors.

In our opinion, the development of model intercomparison studies is a useful way to evaluate the model's performance under different forcing conditions. However, we

also think that these model intercomparison studies must include a large number of modeled sites to reduce the degree of modeling uncertainty. The intercomparison studies carried out by Zhu et al. (2019), Rajesh and Rehana (2021) and Rehana (2019) considered 8, 1 and 1 modeling sites respectively.

In other words, in our opinion, the number of models intercomparison studies:
- i) should increase;
- ii) should cover different regions;
- iii) the number of modeling sites per study should be higher;
- iv) the studies should also focus on the modeling of river water temperature with limiting forcing data.

To clarify the reviewer's concern the following sentence was included in last line of the abstract:

Page 1 – Line 24: "Hopefully, the high number of modeled sections considered in this model intercomparison study and the specific model forcing conditions will help to reduce the overall WT modeling uncertainty."

Referee 2: "The objective of the study is not very clear in the abstract and introduction sections. The authors did not explicitly explain the reason why they selected regions with a lot of missing data as the study domain. What is the novelty of their study compared to other studies that compared model performance?"

RESPONSE: We agree with the reviewer. Reviewer 1 also pointed this out and the following sentences have, therefore, been included in the manuscript abstract and introduction:

Page 1 – Line 11: "Commonly, the WT observed in monitoring stations located near the downstream section of rivers are assumed to be the boundary condition of lake/reservoir water quality models. The main goal of this study is to identify a suitable WT modeling solution for these sections given the scarcity of the forcing datasets."

Page 1 – Line 24: "Hopefully, the high number of modeled sections considered in this model intercomparison study and the specific model forcing conditions will help to reduce the overall WT modeling uncertainty."

Page 3 – Line 91: "Hence, the main objective of this study is to identify a suitable WT modeling solution to improve the lake/reservoir water quality models' boundary

condition. It is important to mention that, for this study, an absence of significant variation between the water quality station observed WT and the WT at the downstream portion of a river was assumed, which coincides with the lake/reservoir water quality model boundary condition."

Page 3 -Line 91: the following sentence:

"It is also important to note that the studies defined to evaluate the performance of different modeling approaches are normally restricted to a very small number of test sites and usually contain a reasonable amount of forcing data (Table 1). Hence, the vital importance of increasing the number of test sites and using a limited amount of forcing data to model river temperatures. This is the primary objective of this study and the methodological approach was, therefore, defined to attempt to answer the following questions:"

was replaced with:

"It is also important to note that the studies defined to evaluate the performance of different modeling approaches are normally restricted to a very small number of test sites and usually contain a reasonable amount of forcing data (Table 1). Hence, the vital importance of increasing the number of test sites and using a limited amount of forcing data to model river water temperatures. This is the primary and innovative objective of this study. The methodological approach was, therefore, defined to attempt to answer the following questions: "

The reviewer is right in their comments regarding the lack of clarity in relation to the method of station selection. We have considered all the stations with available datasets of water temperature and discharge available from the Portuguese Water Resources Information System (SNIRH). To clarify the reviewer's concern the following sentence:

Page 4 – Line 101: "To that end, 83 river sections with different geomorphological, meteorological and hydrological conditions were modeled using five different models, three of which use ML algorithms optimized with a sequential model-based optimization approach: Random Forest (RF); Artificial Neural Network (ANN) and Support Vector Regression (SVR)."

has been replaced with: "To that end, 83 river sections with different geomorphological, meteorological and hydrological conditions were modeled. These stations correspond to all the sections for which the Portuguese Water Resources

Information System (SNIRH) holds WT and discharge datasets, which are also, coincidentally, characterized by 98% missing data. The modeling ensemble includes five different models, three of which use ML algorithms optimized with a sequential model-based optimization approach: Random Forest (RF); Artificial Neural Network (ANN) and Support Vector Regression (SVR). "

Referee 2: I think the authors confused the concept of different data sets in ML and hydrology. In the paper, the validation set was the same as the test set and the validation set, and the training set was the same as the calibration set. Typically, in ML, the study data is divided into three sets (i.e., training, validation and test sets) or two sets (training and test sets). ML models are fitted on the training set to learn the relationship between input and output data. The validation set is used for hyperparameter tuning. The optimal hyperparameters are determined based on validation performance. The validation set is similar to the calibration set in hydrology. Finally, the test set is used to evaluate the ability of the trained ML model to handle previously unobserved data. It is similar to the validation set in hydrology.

RESPONSE: Thank you for pointing this out. We did indeed write 'validation' when we meant 'testing'. Due to the size of the datasets we have only considered a training set and a test set. The optimization algorithm was applied to the training set. We have replaced the word 'validation' with 'testing' throughout the report.

To clarify the reviewer's concern the following sentence was included on:

Page 7 – Line 164: "Due to size of the available datasets the validation phase was not considered."

Referee 2: In the introduction part of Section 3, the authors described the workflow in text. In the study, they first trained ML models at 12 wells, and then apply the 12 models to each well, using the ensemble of the best results obtained across the 12 models per well as the final ML results. A figure showing the workflow would be helpful for understanding.

RESPONSE: We agree with the reviewer. We have included a schematic and simplified representation of the modeling process

Referee 2: In Section 3.5, the authors did not state which data set was used in hyperparameter tuning. How to calculate the algorithm score there?

RESPONSE: Thank you for pointing this out. The following sentences were included to clarify the reviewer concern:

Page 10 – Line 269: "The coefficient of determination ($R^2$) was considered as the algorithm score."

Page 10 – Line 271: "The algorithm was applied to the training data set. Table 4 shows the model parameters and the optimization range."

Referee 2: Too many big tables and too many plots in most figures, which are very distracting. Maybe only show the most important information in the main text, and move the reset to the appendix or supplementary materials.

RESPONSE: We agree with the reviewer and six tables have been transferred to the appendix.

Referee 2: Line 169: "L ≤ 50; 50> L ≤100; 100> L ≤200; L>200" should be "L ≤ 50; 50< L ≤100; 100< L ≤200; L>200"

RESPONSE: Thank you. We have inserted the correction.

Referee 2: Line 294: delete the extra "root".

RESPONSE: Thank you. We have inserted the correction.

Referee 2: Equation 9: Please explain "r".

RESPONSE: Thank you for pointing this out.

Page 12 – Line 308: The following sentence:

"values and $\sigma o$ the standard deviation of the observed values:"

, was replaced with: "values, $\sigma o$ the standard deviation of the observed values and r is the Pearson coefficient:"

Referee 2: Line 326: Please explain "with 3-par". Similar phrases in Table 1.

RESPONSE: Thank you for pointing this out.

The following sentence was included:

Page 10 – Line 254: In this study five versions of this model were considered to model WT. The 3, 4, 5, 7 and 8 parameter versions. Please refer to Toffolon and Piccolroaz (2015) for a full description of each one of the models' parameterizations.

Additionally, in the text 3-par was replaced with 3-parameters.

The following note was included at the end of Table 1:

*The model can be applied with 3, 4, 5, 7 or 8 parameters (3-par; 4-par; 5-par; 7-par and 8-par)

**References**

DeWeber, J. T., and Wagner, T.: A regional neural network ensemble for predicting mean daily river water temperature, J. Hydrol, 517, 187–200, https://doi.org/10.1016/j.jhydrol.2014.05.035, 2014.

Moore, R.D., Nelitz, M., and Parkinson, E.: Empirical modelling of maximum weekly average stream temperature in British Columbia, Canada, to support assessment of fish habitat suitability, Canadian Water Resources Journal / Revue canadienne des ressources hydriques, 38(2), 135-147, https://doi.org/10.1080/07011784.2013.794992, 2013.

Ducharne, A.: Importance of stream temperature to climate change impact on water quality, Hydrol. Earth Syst. Sci., 12, 797– 810, https://doi.org/10.5194/hess-12-797-2008, 2008.

Rajesh, M., and Rehana, S.: Prediction of river water temperature using machine learning algorithms: a tropical river system of India, Journal of Hydroinformatics, 23 (3), 605–626, https://doi.org/10.2166/hydro.2021.121, 2021.

Rehana, S.: River water temperature modelling under climate change using support vector regression, in: Hydrology in a Changing World: Challenges in Modeling, edited by Singh, S. K., and Dhanya, C. T., World Springer, 171–183, https://doi.org/10.1007/978-3-030-02197-9_8., 2019.

Zhu, S., Nyarko, E. K., Hadzima-Nyarko, M., Heddam, S., and Wu, S.: Assessing the performance of a suite of machine learning models for daily river water temperature prediction, PeerJ, 7: e7065, https://doi.org/10.7717/peerj.7065, 2019.

---

## Referee Report (RR1)

The manuscript presents a study that employs five models to predict water temperatures for 83 low order rivers with limited water temperature datasets. The models used include three machine-learning algorithms, the hybrid Air2stream model with all available parameterizations, and a Multiple Regression model. Overall, the authors found the ML techniques exhibited superior predictive performance when coupled with appropriate hyperparameters. Among the models tested, RF demonstrated the best evaluation metrics. The authors suggest that the high number of modelled sections and specific model forcing conditions help to reduce the overall uncertainty in river water temperature modelling. The scientific approach and methods applied in this study are sound, and the results obtained are reliable and reproducible. However, the abstract and introduction are not well-organized and do not clearly state the significance, research gap, and novelty of this study. In addition, some parts of the discussion repeat information presented in the introduction, which should be revised accordingly. I think the present work will be a valuable paper in the field of modelling river temperature with a significant amount of missing values.

**Major comments:**

1. The scope of the main target of this study is formulated in a way that is too narrow. This manuscript presents a study aimed at filling gaps in observed river water temperature datasets to improve boundary conditions for lake/reservoir water quality models. However, this goes far beyond just being the boundary conditions for lake/reservoir models. The authors should consider this more, for instance, it can also be valuable for analyzing seasonal/diurnal trends and biogeochemical processes in rivers based on observation datasets. Furthermore, even for modelling lakes/reservoirs physics/hydrodynamics, inflow temperature is also critical. Therefore, it may be beneficial to at least delete the phrase 'water quality' from the sentence.

2. Besides model performance, the number of the input parameters should also be taken into account when comparing models, for example, in AIC/BIC, model with fewer parameters has a better score. Given the results, I have the feeling that air2stream would be the most favorable alternative, as all the RMSEs were relatively high and similar (all over 3 $^{\circ}$C), and some of the machine learning techniques exhibited signs of overfitting the datasets. However, air2stream needs fewer input predictor variables in comparison to other methods. Moreover, air2stream considers the physical process, which will be more robust.

3. The author reiterated in both the manuscript and response that the models' error can potentially be mitigated through the use of a pre-processing technique, such as SMOGN. Therefore, it would be beneficial to investigate and compare the efficacy of SMOGN against the models' performance on the raw datasets.

4. Some sentences in the discussion repeat the introduction, for example, in line 651 "but can also be associated with the uncertainty caused by the fact that river WT is not only affected by local environmental conditions, but also by upstream conditions" and line 64 "The predictor variables can represent a significant source of uncertainty, as river WT is not only affected by local environmental conditions, but also by upstream conditions (Moore et al., 2005)." To avoid such repetition, it is recommended to consolidate related ideas and present them cohesively throughout the manuscript.

**Minor comments:**

1. Figure 8a: Change the legend "validation" to "testing".

2. Figure 9: Add a legend explaining the different colored dots, and consider plotting a regression line in green that represents the regression during the testing period in Figure 9a and 9c instead of separating Figure 9b and 9d from Figure 9a and 9c.

3. Table 2 and 5: Change "Stdev" to "Standard deviation".

4. Line 158: "The watershed discharge data used to force the models and the water temperature considered for the model's validation are also available from Portuguese Water Resources Information System (SNIRH)." Change the word "validation" to "test".

5. Line 203: "Following this initial analysis, the models (*vide* Sect. 3.1 to 3.6) were applied to each of the 83 input datasets, divided between a training (70% of the entire dataset) and a test dataset (the remaining 30%)." It may not be appropriate to use the terms "training" and "test" for air2stream model. In hydrology, calibration and validation are more accepted terms.

6. Line 289: "The Air2stream model solves a lumped heat-exchange budget between an unknown river section volume, its tributaries, and the atmosphere (Toffolon and Piccolroaz, 2015)." It is suggested to add groundwater term to the sentence describing the air2stream model.

7. Line 305: "In this 305 study five versions of this model were considered to model WT. The 3, 4, 5, 7 and 8 parameter versions." A dot is missing in front of the sentence.

8. Eq (4): Specify the input variables in the equation to improve clarity.

9. Eq (8): Explain what $o_l$ mean?

---

## Referee Report (RR2)

Thanks for addressing the comments and improving the presentation of the manuscript. However, there are still some issues that are unclear. I appreciate that the authors put extra effort into evaluating the performance of over/under-sampling techniques, e.g., SMOGN, which is still not widely used. Here are my comments and questions.

5    1. In the MR methods, can you explain how the model predictor variables, and the number of the variables were determined?

2. Please check the line number in the author's response, for example, I cannot find the sentences in the author's response in line 94.

3. Suggest deleting citations in the conclusion and rephrasing it as something like "Importantly, our study further confirmed the accuracy of the Random Forest model can be significantly improved by using…"

10   4. Consider shortening and focusing the abstract, for example, in line 13: "Therefore, the main goal of this study is to identify a suitable modeling solution for the prediction of river water temperature given the scarcity of the forcing datasets." could be modified to "Therefore, identifying a suitable modeling solution for the prediction of river water temperature with a large scarcity of the forcing datasets is of great importance." Also, consider shortening the description of the methods and results in the abstract.

15

---

## Author Response (AR2)

We sincerely thank the editor and reviewers for taking the time to review our manuscript and providing constructive feedback to improve it.

**RESPONSE TO REFEREE 2**

**Referee 2:** Thanks for addressing my comments and adding a figure to explain their modeling process. The manuscript is easier to read than the previous version. However, its presentation quality still needs to be significantly improved before it is eligible for publication. Here are my comments.
**Response:** Thank you for your comments. We have improved the manuscript presentation quality.

**Referee 2:** Some long sentences need to be rephrased in order to be more understandable. The authors may split them into two shorter ones.
**Response:** Thank you for your comment. The manuscript was reviewed to further improve the clarity of the text.

**Referee 2:** I appreciate it that the authors added Figure 2 to explain their modeling process, but I feel the modeling process is not very clear. The authors may add more information to the caption of Figure 2 to explain their method.
**Response:** We agree with the reviewer. Fig.2 was modified, and the caption was replaced. The modeling process description was also modified as follows:

Line 217 "Given the large number of input datasets and the fact that the optimization process can be very time consuming, the following approach was implemented (Fig. 2):
i)      The 83 stations were ordered as a function of the number of samples (lowest to highest) and were divided into four different classes ($L \leq 50$; $50 < L \leq 100$; $100 < L \leq 200$; $L > 200$). Three stations were selected within each class: 1) the station with the least samples; 2) the station with the most samples; 3) the station with the number of samples that most closely corresponded to the average sample number for each class. The 12 datasets selected corresponded to Stations 1, 7, 12, 13, 22, 29, 30, 46, 59, 60, 73 and 83;
ii)     The ML and TPE algorithms were applied to the 12 datasets. At this stage there were 12 optimized model structures computed with the TPE algorithm for each ML model;
iii) The 12 optimized models obtained for each ML were subsequently applied to the 83 datasets and the best performing model at each station was calculated on the basis of the computed root mean square value (RMSE). Hence, the ensemble

of the best results obtained across the 12 different models for the 83 stations defines the overall ML results."

**Referee 2:** In the result section, the authors mentioned "ensembles" several times, such as ANN ensembles, but did not explain them anywhere. The authors may state that they evaluated the performance of the ensemble of the 12 models and optimal individual models.
**Response:** Thank you. The word "ensembles" was not correct and was replaced with "ensemble". This variable was already described in the manuscript.

Line 227: "Hence, the ensemble of the best results obtained across the 12 different models per station defines the overall ML results."

**Referee 2:** The quality of the figures still needs to be improved.
**Response:** Actioned. All figures have been improved.

**Referee 2:** In the discussion section, the authors mentioned that all models outperformed the others under some criteria. The authors may consider to provide a guideline on the selection of the models.
**Response:** Thank you. The following text was added to the discussion section to clarify the guideline that was considered for the selection of the models: "considering all metrics"

Line: 642 "That said, the RF model ensemble, considering all the metrics, produced the best results for the annual datasets and was the model that provided the greatest contribution in relation to overall ensemble results

Line: 663 "This explains the fact that the best results, considering all the metrics, were obtained with the 3-parameter model"

Specific comments:
**Referee 2:** Line 25-28: The sentence "when the number of predictor variables … (NSE): $0.56 \pm 0.48$)" is not clear to me. I am not sure what the authors mean here.
**Response:** Thank you. To clarify the reviewer's concern the following sentence has been modified:

This sentence:
"In general terms, the results of the study demonstrate the vital importance of hyperparameter optimization and suggest that, from a practical modeling perspective, when the number of predictor variables and observed river WT

values are limited, the application of all the models considered in this study is relevant (models ensemble mean annual – Root mean square error (RMSE): 2.75 ºC ± 1.00; Nash-Sutcliffe efficiency (NSE): 0.56 ± 0.48). Therefore, the datasets gaps can be filled with the best model of the ensemble approach."

Was replaced with: "In general terms, the results of the study demonstrate the vital importance of hyperparameter optimization and suggest that, from a practical modeling perspective, when the number of predictor variables and observed river water temperature values are limited, the application of all the models considered in this study is crucial. Basically, all the models tested proved to be the best for at least one station. The Root Mean Square Error (RMSE) and the Nash-Sutcliffe efficiency (NSE) values obtained for the ensemble of all model results was 2.75ºC ± 1.00 and 0.56ºC ± 0.48, respectively."

**Referee 2:** Line 97-98: It seems that the sentence "Hence, the vital importance …river temperatures" misses the verb.
**Response:** We have reviewed the sentence and we think that it is correct.

**Referee 2:** Line 155: What do "sections" mean here? Do they mean stations?
**Response:** The reviewer is right. Thank you. "sections" was replaced with "stations".

**Referee 2:** Line 182: Should it be "were computed"?
**Response:** The reviewer is right. Thank you.

Line 231. "where computed" was replaced with " were computed".

**Referee 2:** Table 3: Replace "ML" to "MR". Why does Air2stream only have 2 input variables?
**Response:** Corrected.

**Referee 2:** Line 207: Could you explain which 12 datasets were used for optimization?
**Response:** Yes. The following sentence has been included in the manuscript.

Line 217: "). Given the large number of input datasets and the fact that the optimization process can be very time consuming, the following approach was implemented (Fig. 2):
i)      The 83 stations were ordered as a function of the number of samples (lowest to highest) and were divided into four different classes (L ≤ 50; 50< L

≤100; 100< L ≤200; L>200). Three stations were selected within each class: 1) the station with the least samples; 2) the station with the most samples; 3) the station with the number of samples that most closely corresponded to the average sample number for each class. The 12 datasets selected corresponded to Stations 1, 7, 12, 13, 22, 29, 30, 46, 59, 60, 73 and 83;"

**Referee 2:** Line 261: I think here should be testing dataset.
**Response:** This sentence is correct. "The model has no capacity to provide information on energy flux mechanisms within the river and has a tendency to overfit the training dataset, thereby considerably diminishing the model's ability to generalize the features or patterns present in the training dataset (Srivastava et al., 2014)."

**Referee 2:** Line 306: Please specify which disadvantages of the Gaussian Process the TPE algorithm fixes.
**Response: 2.** Thank you. The following sentence has been included:

"It can be difficult to select the right hyperparameters for GP with EI due to the many different Kernel types associated with this process. TPE uses simpler Kernels as a building block, which facilitates hyperparameter selection. Furthermore, TPE is faster than GP with EI when the number of hyperparameters increases."

**Referee 2:** Line 313: Table 3 does not provide the corresponding optimization range.
**Response:** The reference in the text was not correct. The optimization range of the models' parameters is provided in Table A1.

"Table A1 shows the models parameters and the corresponding optimization range."

**Referee 2:** Table 4: Please provide unit to the metrics.
**Response:** Corrected. Thank you.

**Referee 2:** Figure 3: What does the x axis mean?
**Response:** Thank you for pointing this out. It means the number of bins.
The figure has been corrected.

**Referee 2:** Figure 7: I believe here is MAE in the figure titles.
**Response:** Thank you. The figure has been corrected.

**Referee 2:** Figure 9: Please explain Ct in the caption.

**Response:** Corrected. Thank you.

**RESPONSE TO REFEREE 3**

The manuscript presents a study that employs five models to predict water temperatures for 83 low order rivers with limited water temperature datasets. The models used include three machine-learning algorithms, the hybrid Air2stream model with all available parameterizations, and a Multiple Regression model. Overall, the authors found the ML techniques exhibited superior predictive performance when coupled with appropriate hyperparameters. Among the models tested, RF demonstrated the best evaluation metrics. The authors suggest that the high number of modelled sections and specific model forcing conditions help to reduce the overall uncertainty in river water temperature modelling. The scientific approach and methods applied in this study are sound, and the results obtained are reliable and reproducible. However, the abstract and introduction are not well-organized and do not clearly state the significance, research gap, and novelty of this study. In addition, some parts of the discussion repeat information presented in the introduction, which should be revised accordingly. I think the present work will be a valuable paper in the field of modelling river temperature with a significant amount of missing values.

**Response:** We thank the reviewer again for the positive feedback. We agree with the reviewer. The abstract, introduction and discussion sections were improved.

Major comments:

**Referee 3**: 1. The scope of the main target of this study is formulated in a way that is too narrow. This manuscript presents a study aimed at filling gaps in observed river water temperature datasets to improve boundary conditions for lake/reservoir water quality models. However, this goes far beyond just being the boundary conditions for lake/reservoir models. The authors should consider this more, for instance, it can also be valuable for analyzing seasonal/diurnal trends and biogeochemical processes in rivers based on observation datasets. Furthermore, even for modelling lakes/reservoirs physics/hydrodynamics, inflow temperature is also critical. Therefore, it may be beneficial to at least delete the phrase 'water quality' from the sentence.

**Response:** We agree with the reviewer. The following sentences have been included in the manuscript:

Line 10 – "Water temperature datasets for low order rivers are often in short supply, leaving environmental modelers with the challenge of extracting as much information as possible from existing datasets, usually without the use of physically based models, due to the significant amount of data required (e.g., river morphology, degree of shading, wind velocity)."

Line 94 – "Therefore, the main objective of this study is to identify a suitable WT modeling solution for rivers with limiting forcing data. Improving this type of solution would deliver potential benefits for a wide range of environmental modeling applications, such as the analysis of seasonal/diurnal trends and biogeochemical processes in rivers based on observation datasets and the improvement of lake/reservoir water quality model boundary conditions."

**Referee 3**: 2. Besides model performance, the number of the input parameters should also be taken into account when comparing models, for example, in AIC/BIC, model with fewer parameters has a better score. Given the results, I have the feeling that air2stream would be the most favorable alternative, as all the RMSEs were relatively high and similar (all over 3 oC), and some of the machine learning techniques exhibited signs of overfitting the datasets. However, air2stream needs fewer input predictor variables in comparison to other methods. Moreover, air2stream considers the physical process, which will be more robust.

**Response:** Thank you for your comment. As the reviewer will be aware, machine learning algorithms do not allow the computation of an AIC or BIC. The majority of these methods are neither likelihood-based, nor can one readily account for model complexity, because the number of parameters does not reflect the effective degrees of freedom (Hauenstein, 2017). We agree with the reviewer that considering the AIC/BIC definition the Air2stream model would produce a better score. We also think that a simple linear regression would also have produced a better score considering the penalization that the AIC/BIC equations give to the number of models parameters and the complexity of ML models. However, we do not see this as a fair comparison. One of the main conclusions drawn from this study is that, from a practical perspective, all models should be applied, as all models performed best for at least one station. The Air2stream considers the physical process, however, it also has some relevant simplifications. Overall, by considering a metric that is easily interpretable, such as the MAE, and 83 testing sites, the Air2stream performance was not better than the ML models.

From a practical perspective this is also a measure of model robustness. When describing nonlinear correlations, ML models have frequently performed as well or better than physical-based models with less input data (Virro et al., 2022).

**Referee 3**: 3. The author reiterated in both the manuscript and response that the models' error can potentially be mitigated through the use of a pre-processing technique, such as SMOGN. Therefore, it would be beneficial to investigate and compare the efficacy of SMOGN against the models' performance on the raw datasets.

**Response:** Thank you for this suggestion. As the reviewer mentioned we were trying to provide an additional approach to further improve the modeling results. We also mentioned that: "This algorithm was not implemented, because the user must assign more importance to the predictive performance obtained for some poorly represented ranges, in comparison to other more frequent ranges. In our opinion, this process needs to be driven by the water quality model temperature calibration process. Hence, we have chosen to preserve the original datasets and to evaluate the model's performance over the raw datasets."

However, we think that a reasonable implementation of SMOGN and the availability of the code can be beneficial for the manuscript and for the readers. We found a balanced way to implement SMOGN, considering:
1) that this process can be very time-consuming;
2) the initial manuscript structure and methodological approach.

We have generated 100 synthetic training datasets for each of the 12 raw datasets that were initially considered to define the 12 optimized models. Hence, for each of the 12 datasets, the best model (random forest) was optimized and trained with 100 different training datasets for each of the 12 stations. The methodology and discussion sections have been updated and a new section has been included to describe the modeling results (See Section 4.4). The SMOGN code has been added to the code and data repository (Almeida and Coelho, 2023).

**Referee 3**: 4. Some sentences in the discussion repeat the introduction, for example, in line 651 "but can also be associated with the uncertainty caused by the fact that river WT is not only affected by local environmental conditions, but also by upstream conditions" and line 64 "The predictor variables can represent a significant source of uncertainty, as river WT is not only affected by local

environmental conditions, but also by upstream conditions (Moore et al., 2005)."
To avoid such repetition, it is recommended to consolidate related ideas and present them cohesively throughout the manuscript.
**Response:** Thank you, the manuscript text has been reviewed.

Minor comments:
**Referee 3**: 1. Figure 8a: Change the legend "validation" to "testing".
**Response:** Actioned.

**Referee 3:** 2. Figure 9: Add a legend explaining the different colored dots, and consider plotting a regression line in green that represents the regression during the testing period in Figure 9a and 9c instead of separating Figure 9b and 9d from Figure 9a and 9c.
**Response:** Actioned.

**Referee 3:** 3. Table 2 and 5: Change "Stdev" to "Standard deviation".
**Response:** Corrected. Thank you.

**Referee 3:** 4. Line 158: "The watershed discharge data used to force the models and the water temperature considered for the model's validation are also available from Portuguese Water Resources Information System (SNIRH)." Change the word "validation" to "test".
**Response:** Corrected. Thank you.

**Referee 3:** 5. Line 203: "Following this initial analysis, the models (vide Sect. 3.1 to 3.6) were applied to each of the 83 input datasets, divided between a training (70% of the entire dataset) and a test dataset (the remaining 30%)." It may not be appropriate to use the terms "training" and "test" for air2stream model. In hydrology, calibration and validation are more accepted terms.
**Response:** Thank you. We agree with the reviewer and the following sentence has been included:

Line 214 "It should be noted that, in the case of the Air2stream model, 70% of the initial dataset corresponds to the calibration dataset and the remaining 30% to the validation dataset."

**Referee 3:** 6. Line 289: "The Air2stream model solves a lumped heat-exchange budget between an unknown river section volume, its tributaries, and the

atmosphere (Toffolon and Piccolroaz, 2015)." It is suggested to add groundwater term to the sentence describing the air2stream model.

**Response:** Thank you for the suggestion, the sentence has been modified.

Line 281. "The Air2stream model solves a lumped heat-exchange budget between an unknown river section volume, its tributaries, groundwater, and the atmosphere (Toffolon and Piccolroaz, 2015)."

**Referee 3:** 7. Line 305: "In this study five versions of this model were considered to model WT. The 3, 4, 5, 7 and 8 parameter versions." A dot is missing in front of the sentence.

**Response:** Corrected. Thank you.

**Referee 3:** 8. Eq (4): Specify the input variables in the equation to improve clarity.
**Response:** Actioned.

**Referee 3:** 9. Eq (8): Explain what ol mean?
**Response:** Thank you. ol has been replaced with $\overline{o}$, which is the observed values mean defined in the original document.

**References**

Almeida, M.C. and Coelho, P.S.: mcvta/WaterPythonTemp: Release 0.2.0, Zenodo [code]. https://doi.org/10.5281/zenodo.7870379, 2023

Hauenstein, S., Wood, S.N. and Dormann C.F..: Computing AIC for black-box models using generalized degrees of freedom: A comparison with cross-validation, Communications in Statistics - Simulation and Computation, 47:5, 1382-1396, DOI: 10.1080/03610918.2017.1315728, 2018.

Virro H, Kmoch A, Vainu M, Uuemaa, E.: Random forest-based modeling of stream nutrients at national level in a data-scarce region. Science of the Total Environment 840: 156613. Volume 840, https://doi.org/10.1016/j.scitotenv.2022.156613, 2022.

---

## Author Response (AR3)

**We sincerely thank the editor and reviewers for taking the time to review our manuscript and providing constructive feedback to improve it.**

**Referee:** Thanks for addressing the comments and improving the presentation of the manuscript. However, there are still some issues that are unclear. I appreciate that the authors put extra effort into evaluating the performance of over/under-sampling techniques, e.g., SMOGN, which is still not widely used. Here are my comments and questions.

**Response:** Thank you for the time spent and for the thoughtful comments and suggestions towards improving our manuscript.

**Referee:** 1. In the MR methods, can you explain how the model predictor variables, and the number of the variables were determined?

**Response:** Thank you for your comment. Please note that this explanation can already be found in the Methods section, line 191.

Line 191: "Initially the model predictors were selected on the basis of their availability and the results obtained with other studies (e.g., Zhu et al., 2019c; Feigl et al., 2021)."

**Referee:** 2. Please check the line number in the author's response, for example, I cannot find the sentences in the author's response in line 94.

**Response:** Thank you. We have checked the line numbers. The sentence is in line 97.

**Referee:** 3. Suggest deleting citations in the conclusion and rephrasing it as something like "Importantly, our study further confirmed the accuracy of the Random Forest model can be significantly improved…"

**Response:** Thank you for your comment. The citation was removed, and the sentence was rephrased as follows:

**Line 717:** "Importantly, our study further confirmed the accuracy of the Random Forest can be significantly improved by the generation of synthetic samples to some poorly represented ranges within the training datasets by applying an over/undersampling technique."

**Referee:** 4. Consider shortening and focusing the abstract, for example, in line 13: "Therefore, the main goal of this study is to identify a suitable modeling solution for the prediction of river water temperature given the scarcity of the forcing datasets." could be modified to "Therefore, identifying a suitable modeling solution for the prediction of river water temperature with a large scarcity of the forcing datasets is of great importance." Also, consider shortening the description of the methods and results in the abstract.

**Response:** Thank you for your comment. We have included the reviewer suggestion and we have shortened the abstract as much as possible.